# Phylogenomic and Evolutionary Insights into Lipoprotein Lipase (LPL) Genes in Tambaqui: Gene Duplication, Tissue-Specific Expression and Physiological Implications

**DOI:** 10.3390/genes16050548

**Published:** 2025-04-30

**Authors:** Rômulo Veiga Paixão, Izabel Correa Bandeira, Vanessa Ribeiro Reis, Gilvan Ferreira da Silva, Fernanda Loureiro de Almeida O’Sullivan, Eduardo Sousa Varela

**Affiliations:** 1Instituto Federal de Educação, Ciência e Tecnologia do Amazonas (IFAM), Campus Presidente Figueiredo, Av. da Onça-Pintada, S/N-Galo da Serra, Presidente Figueiredo CEP 69735-000, AM, Brazil; 2Embrapa Pesca e Aquicultura, Palmas, TO, Brazil, Av. NS 10, Cruzamento com a Av. LO 18 Sentido Norte Loteamento-Água Fria, Palmas CEP 77008-900, TO, Brazil; fernanda.almeida@embrapa.br; 3Embrapa Amazônia Ocidental, Rodovia AM-010, Km 29, Caixa Postal 319, Manaus CEP 69010-790, AM, Brazil; izacbandeira@gmail.com (I.C.B.); reis.vanessar@gmail.com (V.R.R.); gilvan.silva@embrapa.br (G.F.d.S.)

**Keywords:** lipid metabolism, lipoproteins, lipase, gene duplication, comparative genomics, gene expression, neotropical fish

## Abstract

Background/Objectives: Lipoprotein lipase (LPL) is a key enzyme in lipid metabolism, crucial for the hydrolysis of triglycerides in lipoproteins and maintaining lipid homeostasis in vertebrates. This study aims to characterize the lipoprotein lipase genes in the tambaqui (*Colossoma macropomum*) genome, investigating their evolutionary history from a phylogenomic perspective. Methods: Phylogenetic and syntenic analyses were used to identify the *lpl* gene copies in the tambaqui genome and expression patterns were examined across different tissues. A comparative analysis with *lpl* genes from other vertebrates was also conducted to assess evolutionary relationships and functional diversification. Results: We identified three *lpl* gene copies in the tambaqui genome: *lpl1a*, *lpl1b*, and the lesser-known member of the lipoprotein lipase subfamily, *lpl2a*. These proteins possess conserved sites essential for lipoprotein lipase function, with variations that may affect their physicochemical properties and lipolytic activity. Key amino acid variations, such as in the lid region and glycosylation sites, were observed among orthologs. Gene expression analysis showed high *lpl1a* and *lpl2a* expression in the liver, and *lpl1b* expression in the gonads, suggesting tissue-specific roles. Comparative analysis revealed distinct expression patterns among teleost fish, with tambaqui exhibiting a unique profile consistent with its migratory lifestyle and varied diet. Conclusions: This study offers new insights into the evolution and functional diversification of lipoprotein lipases in vertebrates, highlighting the complexity of lipid metabolism in fish. These findings contribute to understanding the adaptability of teleost fish to diverse environments and lay the foundation for future research in lipid metabolism regulation, including Neotropical species, with potential applications in aquaculture and conservation.

## 1. Introduction

Lipoprotein lipase (LPL) is a key enzyme involved in lipid metabolism, playing a crucial role in the hydrolysis of triglycerides from circulating lipoproteins, such as chylomicrons and VLDL, and their uptake by tissues [1]. This enzyme belongs to the vascular lipase gene family, which encompasses hepatic (LIPC) and endothelial lipases (LIPG), sharing significant sequence similarities [2].

In mammals, LPL is encoded uniquely by the *lpl* gene [3,4,5], and the structures within the mammalian neutral lipase family have been determined by molecular modeling of human LPL [6]. There are three main structural domains associated with LPL, including an N-terminal domain which contains the crucial catalytic triad of amino acids (serine, aspartate, and histidine), fundamental to the enzyme’s catalytic activity; a lid domain which is a flexible structure that moves to cover or uncover the active site of the enzyme, essential for regulating access to the enzyme’s active site and contributes to substrate specificity such as triglycerides and phosphoglycerides; and a C-terminal or ‘plat’ domain which is involved in lipid binding, helping to anchor the enzyme to the lipid interface or membrane surfaces [1,2,6,7].

Extensive research has elucidated the critical functions of LPL in mammalian lipid metabolism, with implications for understanding metabolic diseases, cardiovascular health, and energy homeostasis [1,8,9,10,11]. Advances in mammalian LPL studies have highlighted its regulatory mechanisms, tissue-specific expressions, and interactions with other metabolic pathways, offering comprehensive insights into its role and potential therapeutic targets [12,13,14,15,16]. Compared with the extensive research in mammals, only a few investigations on Lpl have been performed in fishes.

There are notable differences in lipid storage sites between fish and mammals, with tissue distribution of lipid metabolic enzymes reflecting their activities [17]. In fish species, lipids are stored in different parts of the body: visceral (including hepatic), subcutaneous, and intramuscular adipose tissues [18]. The enzymes LPL and Lpl, which are involved in lipid metabolism, can be found in various tissues of mammals [5,19] and fishes [20,21], respectively. However, researchers have reported that *lpl* in fish species is mainly expressed in the liver and muscle [22,23], whereas in mammals it is not detected in the adult liver [24,25].

Lpl is essential for the mobilization and distribution of lipids in fishes, which are important energy sources for growth, reproduction, and adaptation to different environmental conditions [26]. The study of Lpl in fish has gained increasing attention due to its potential implications for aquaculture, as lipid metabolism is closely related to fish growth, feed efficiency, and flesh quality [27]. Dietary lipids significantly affected *lpl* mRNA levels in dark barbel catfish (*Pelteobagrus vachelli*) larvae during early ontogeny [28]. In red sea bream (*Pagrus major*), dietary fatty acids affected *lpl* mRNA expression levels in visceral adipose tissue and liver of fed and starved fish [26]. In Atlantic salmon (*Salmo salar* L.), hepatic *lpl* expression levels were 3.8-fold higher in fish fed tetradecylthioacetic acid compared to those fed fish oil, although muscle *lpl* expression levels were not significantly different between the two groups [29]. In tilapia (*Oreochromis niloticus*), high dietary lipids induced expression of hepatic *lpl* [27,30]. In the Dabry’s sturgeon *Acipenser dobryanus*, the expression levels of *LPL* were significantly affected by feeding frequency [21]. In grouper (*Epinephelus coioides*), the replacement of fish oil (FO) with palm oil (PO) in the diet led to changes in the mRNA expression of genes related to hepatic lipid metabolism, including *lpl* [31].

To date, research has primarily focused on a single *lpl* gene in fishes. However, one study reported a second type in cutthroat trout *Oncorhynchus clarki*, named *lpl2-like*, predominantly expressed in the granulosa cells of ovarian follicles, with features similar to LPLs/Lpls of other species, including several conserved structural and functional domains [32]. Teleost fishes have undergone a whole-genome duplication event (3R) during their evolution, resulting in the presence of multiple copies of various genes [33], including *lpl*. The retention and divergence of these duplicated genes have been proposed to play a role in the adaptation and diversification of teleosts [34]. However, the evolutionary history and functional implications of *lpl* gene duplication in teleosts remain poorly understood, especially for Neotropical freshwater fishes where research about lipases has so far been limited to the enzymatic level [35].

Tambaqui (*C. macropomum*) is a Neotropical freshwater fish species of significant economic importance in South American aquaculture [36]. As an omnivorous fish with a high capacity for lipid accumulation [37,38], tambaqui represents an interesting model to study the evolution and function of *lpl* in teleosts, and more specifically in Neotropical species. Understanding the molecular basis of lipid metabolism in tambaqui may provide valuable insights for the improvement of its aquaculture production and the development of sustainable feeding strategies. In this context, this study aims to fill this gap by identifying and characterizing *lpl* genes in tambaqui and comparing their sequences and expression patterns with those of other teleost species. Using bioinformatics tools and gene expression techniques, we sought to better understand the functional and evolutionary diversity of these genes, as well as their implications for tambaqui physiology.

## 2. Materials and Methods

### 2.1. In Silico Analysis

#### 2.1.1. Identification of *lpl* Sequences in Tambaqui and Other Teleost

Tambaqui *lpl* genes were identified using BLAST (online version, accessed on 1 July 2024) searches against juvenile trunks, ovary, and testis RNAseq data (Bioproject: PRJEB40318), and the *C. macropomum* genome assembly from NCBI [39,40,41]. The retrieved sequences were manually curated using the Unipro UGENE v.35.1 software to construct the predicted CDS, which were subsequently used to design species-specific primers for real-time quantitative PCR (qPCR) and to deduce the amino acid sequences.

Non-redundant protein sequence databases for several vertebrate genomes were examined using the BLASTP algorithm. Teleost representatives include the superorder Elopomorpha (*Megalops cyprinoides*, and *Anguilla anguilla*), Osteoglossomorpha (*Sclerophages formosus*, and *Paramormyrops kingsleyae*), Otocephala (*Clupea harengus*, *Chanos chanos*, *Danio rerio*, *Cyprinus carpio*, *Electrophorus electricus*, *Ictalurus punctatus*, *Pangasianodon hypophthalmus*, *Pygocentrus nattereri*, and *Astyanax mexicanus*), and Eutelostei (*Gasterosteus aculeatus*, *Takifugu rubripes*, *Dicentrarchus labrax*, *Esox lucius*, *O. niloticus*, *Oryzias latipes*, *Gadus morhua*, *S. salar*, and *Onchorhynchus mykiss*). We also included non-teleost actinopterygian such as *Lepisosteus oculatus*, and *Erpetoichthys calabaricus*, some sarcopterygians (*Latimeria chalumnae*, *Homo sapiens*, *Gallus gallus*, *Xenopus tropicalis*, and *Podarcis muralis*), and one representative of Chondrichthyes (*Callorhinchus milii*). This procedure produced multiple BLAST hits for LPL amino acid sequences in each of the protein databases, which were individually examined and retained in FASTA format for phylogenetic analysis.

#### 2.1.2. Phylogenetic Analysis of *lpl* Genes

Multiple sequence alignments were performed using MUSCLE v3.7, included in MEGA7. Phylogenetic relationships were estimated using maximum likelihood and Bayesian approaches. The best-fitting model for amino acid substitution matrix (JTT+F+I+G4) was selected based on the proposed model tool in IQ-Tree 2.0, which was also used for the maximum likelihood analysis to obtain the best tree. Node support was assessed with 10,000 bootstrap pseudoreplicates using the ultrafast routine. Bayesian searches were conducted in MrBayes v.3.1.248, with two independent runs of six simultaneous chains for 1,000,000 generations, and sampling every 1000 generations using default priors.

#### 2.1.3. Synteny Analysis of *lpl* Gene Copies Among Teleosts

The Genomicus vertebrate server [42], synchronized with Ensembl releases, was used for chromosomal localization of lipoprotein lipase genes and their neighboring genes which appear in conserved position between the phylogenetic representative fish species for synteny analysis. The genomic regions (44 neighboring genes) of *lpl* and its copies in *C. macropomum* were identified and manually compared using the NCBI scaffold annotations (NW_023494799.1 and NW_023494807.1) against those of other representative species.

#### 2.1.4. Predicted Structures and Properties of Tambaqui Lipoprotein Lipases

To predict the structures and properties of lipoprotein lipases in tambaqui (*C. macropomum*), we performed three different sequence alignments, each focused on a different Lpl variant. Human LPL was used as a reference to identify sites conserved and validated in previous studies [1,2,43]. The physicochemical properties, including molecular weight, isoelectric point (pI) and stability of the Lpls, were predicted using the ProtParam tool available on the ExPASy (Expert Protein Analysis System) server https://web.expasy.org/protparam/ (accessed on 7 March 2025). The NetNGlyc 1.0 Server was used to predict potential N-glycosylation sites for vertebrate LPL proteins http://www.cbs.dtu.dk/services/NetNGlyc/ (accessed on 16 July 2024).

### 2.2. Gene Expression Analysis of Tambaqui lpl Gene Copies

#### 2.2.1. Sample Collection and RNA Extraction

Tambaqui (*C. macropomum*) tissue samples from the liver, gonads, brain, intestine, stomach, pyloric caeca, heart, and muscle were collected from adult specimens (*n* = 3), maintained under controlled conditions in an aquaculture laboratory at Embrapa Amazonia Ocidental. For sampling, fish were deeply sedated with 250 mg L^−1^ benzocaine before being euthanized by cranial perforation. Total RNA was extracted from tissues using TRIzol reagent (Invitrogen, Carlsbad, CA, USA) according to the manufacturer’s instructions. RNA purity and concentration were assessed using a NanoDrop spectrophotometer (Thermo Fisher Scientific, Wilmington, DE, USA). RNA integrity was verified by 1% agarose gel electrophoresis.

#### 2.2.2. cDNA Synthesis and qPCR

Total RNA samples were treated with TURBO™ DNase (Invitrogen™, Waltham, MA, USA) to remove any possible genomic DNA residues. The concentration and integrity of the RNA were assessed by spectrophotometry (Nanodrop 1000; Thermo Scientific, Waltham, MA, USA) and in agarose gel electrophoresis (1.5%), respectively. Only samples with a 260/280 ratio between 1.8 and 2.1 were used for cDNA synthesis, which was performed using High-Capacity cDNA Reverse Transcription Kit (Applied Biosystems, Foster City, CA, USA), according to the manufacturer’s protocol. The synthesis reaction was carried out in a thermocycler (Applied Biosystems™) with the following program: 25 °C for 10 min, 37 °C for 120 min and 85 °C for 5 min. The cDNAs were used as a template to amplify and quantify the transcript levels of *lpl* gene copies. All qPCR primers (Appendix A) were designed using the Integrated DNA Technologies (IDT) tools https://www.idtdna.com (accessed on 10 November 2023, based on tambaqui genomic nucleotide sequences. Amplification efficiency for each primer set was calculated from five-point, 1:4 serial dilution curves, using pooled liver cDNA for *lpl1a* and *lpl2a*, and pooled ovary and testis cDNA for *lpl1b*. The RT-qPCR assays showed high linearity and efficiency for all target genes, with amplification efficiencies of 101.39% (*lpl1a*), 97.75% (*lpl1b*), 95.99% (*lpl2a*), and 100.88% (*β-actin*). Gene expression of *lpl* gene copies was quantified by real-time PCR (qPCR) using the 7500 Fast Real-Time PCR System v2.3 (Applied Biosystems™). Reactions were performed in a final volume of 10 µL containing 5 µL of SYBR^®^ Green Master Mix (Applied Biosystems™), 0.5 µL of each primer (200 nM), 1 µL of cDNA, and 3.5 µL of nuclease-free water. Cycling conditions were as follows: 95 °C for 10 min followed by 40 cycles of 95 °C for 15 s and 60 °C for 1 min. The specificity of the amplified products was confirmed by dissociation curve analysis. Gene expression data were normalized using the *β-actin* reference gene [44] and analyzed by the method 2^−∆∆Ct^ [45]. For this, the Delta Ct of each sample was calibrated against the average Delta Ct of the tissue with the lowest Delta Ct for each gene.

#### 2.2.3. Statistical Analysis

To compare gene expression between tissues, we used the average ΔΔCt, where ΔΔCt = ΔCt (Sample) − ΔCt (Control Average). Statistical analysis was performed using GraphPad Prism software (v. 8.0.1). First, we identified outliers using the ROUT method (Q = 1%). Next, we conducted the Shapiro–Wilk test for normality to assess the distribution of the data. Tissues that passed the normality test were then compared using Welch’s *t*-test for independent samples, with significance defined at *p* < 0.05. Results are presented as bar graphs, showing the mean ± SD of 2^−ΔΔCt^ values.

### 2.3. Comparative Gene Expression Analysis of lpl Gene Copies Across Teleosts

The average depth values for each *lpl* copy were obtained from the RNAseq libraries available in the Phylofish database https://web.archive.org/web/20250212070258/http://phylofish.sigenae.org/index.html (accessed on 17 July 2024) using the TBLASTN algorithm with protein sequences as queries (Appendix A). These values were used qualitatively to visualize tissue-specific expression patterns, as sequencing depth across species and tissues was standardized in PhyloFish [34]; nonetheless, differences in transcriptome completeness and expression dynamics may still limit direct quantitative comparisons. Representative libraries from various tissues, including liver, kidneys, gonads, heart, bones, brain, gills, and intestine, were selected from a non-teleost fish (*L. oculatus*) and multiple teleost fish species, including those from the Early branching teleost clade (*Osteoglossum bicirrhosum*), Otocephala (*Alosa alosa*, *D. rerio*, *P. hypophthalmus*, *A. mexicanus* [both cave and surface forms]), and the Euteleost clade (*Plecoglossus altivelis*, *G. morhua*, *O. latipes*, and *Perca fluviatilis*). The results were presented as a heatmap, displaying the average depth values for each tissue and species. To facilitate visualization and interpretation, a color gradient was applied, with more intense colors indicating higher expression levels of the gene of interest, and softer colors representing lower expression levels. All data used in this study are available in the Phylofish database, ensuring transparency and the possibility of replicating the analyses performed.

## 3. Results

### 3.1. Tambaqui lpl Genes

Three copies of *lpl* were detected in the tambaqui genome (Figure 1). Initially, the genes were annotated in GenBank as *lpla* and *LOC118818927* (*lpl-like*), which are located adjacent to each other on chromosome LG 19, while *LOC118797146* is located on chromosome LG 23. All three genes are organized into ten exons, with coding sequences (CDS) of 1524, 1596, and 1521 bp, respectively, resulting in predicted proteins of 507, 531, and 506 amino acids. To facilitate reference throughout the manuscript, we refer to *LOC118818927* and *LOC118797146* as *lpl2a* and *lpl1b*, respectively, based on their genomic position and preliminary sequence similarity to known *lpl* paralogs. The phylogenetic and synteny-based rationale for this nomenclature is detailed in Section 3.2 and Section 3.3. Thus, tambaqui Lpl1a shares 63.29% identity with Lpl1b and 51.43% with Lpl2a, while Lpl1b and Lpl2a share 47.61% identity.

### 3.2. Phylogenetic Analysis of Teleost’s lpl Gene Copies

Seventy-eight sequences covering LPL and LPL-like proteins were recovered from 29 vertebrate genomes (Appendix A) to analyze phylogenetic relationships and predict ancestral gene duplications and losses among vertebrates. The phylogram revealed distinct clusters of vertebrate LPL sequences, which were referred to as LPL1 (LPL) and LPL2 (LPL-like) (*p* = 0.9). Each clade grouped the sequences into three main vertebrate groups: chondrichthyans, sarcopterygians, and actinopterygians (the latter encompassing non-teleosts, Early branching teleosts, and clupeocephalans) (Figure 2).

LPL1 and LPL2 sequences were both identified in the Elephant shark (*C. milii*), where they are located on the same chromosome, and in the Coelacanth (*L. chalumnae*), as well as in all non-teleost species analyzed.

In the clade of sarcopterygian LPL sequences, branches were consistent with the phylogenetic relationships within this group: the coelacanth was positioned at the base, followed by amphibians and then amniotes, with two groups (sauropsids, *G. gallus*, reptiles *Podarcis muralis*, and mammals *H. sapiens*) included in the amniote branch. In the actinopterygians clade, a single copy (LPL) was found in non-teleost fish, positioned at the base of the clade that groups all teleost sequences (*p* = 1.0).

In the teleost clade, Lpl1 sequences were grouped into two sister clades: Lpl1a and Lpl1b, consistent with a duplication event that may correspond to the teleost-specific 3R (*p* = 1.0). Lpl1a occurs in all teleost fish analyzed, while the Lpl1b paralog is restricted to Early branch teleosts, with the exception of *S. formosus*. It is also found in Otocephala species, including tambaqui, as well as in salmonids such as *E. lucius*, *S. salar*, and *O. mykiss*, and in a few euteleosts such as *G. morhua* and *Myripristis murdjan*.

In the case of tambaqui, the Lpl1a and Lpl1b sequences clustered within the LPL1 clade, while Lpl2a grouped within the LPL2 clade. Additional paralogs were found in *C. carpio* (Lpl1a1, Lpl1a2, and Lpl1a3) and *O. mykiss* (Lpl1a1 and Lpl1a2) which is consistent with lineage-specific whole genome duplications (4R).

Furthermore, the phylogenomic analysis revealed that, in the LPL2 clade, the ray-finned fish sequences group together in a clade adjacent to the only lobe-finned fish representative, the coelacanth *L. chalumnae* (*p* = 1.0). The unique LPL2 sequence from non-teleost actinopterygians (chondrosteans and holosteans) was positioned at the base, adjacent to a clade containing all teleost sequences (*p* = 1.0). Additionally, in the teleost clade, the unique Lpl2 sequences detected in Early branching teleosts were positioned at the base of the clade containing all unique Lpl2 sequences from other teleosts (Clupeocephala), with the exception of *P. kingsleyae*, which has an additional sequence located outside this major clade (*p* = 1.0).

### 3.3. Synteny Analysis

The synteny map (Figure 3) indicates that the *lpl1* and *lpl2* genes are paralogs resulting from a duplication event that occurred before the divergence of Sarcopterygii and Actinopterygii, as evidenced by the presence of both genes in the Elephant shark (*C. milii*). These genes are located side by side in the analyzed genomes, suggesting a tandem duplication that has been conserved among vertebrates. The genomic region of the Spotted gar (*L. oculatus*), a non-teleost actinopterygian, was used as a reference due to its basal evolutionary position, and the conservation of syntenic blocks between the spotted gar and other species, except tetrapods, indicates significant preservation of these genomic regions throughout evolution.

In all studied teleosts, this *lpl1-lpl2* genomic region was duplicated in two paralogons, in agreement with the 3R. In the Early branching teleost group of osteoglossomorphs, both genomic regions of *P. kingsleyae* contain *lpl1a–lpl2a* (referred to as paralogon A) and *lpl1b-lpl2b* (referred to as paralogon B) located on different chromosomes. In contrast, the Asian bonytongue (*S. formosus*), only retains one of the duplicated genomic regions, which contains *lpl1a*-*lpl2a*. In Clupeocephala, including all Otocephala fishes and some euteleosts such as Northern pike (*E. lucius*) and Cod (*G. morhua*), the duplicated genomic regions contain the *lpl1a-lpl2a*, while the other paralogon genomic region contains only *lpl1b*. However, in euteleost fishes such as Nile tilapia (*O. niloticus*) and Japanese medaka (*O. latipes*), no *lpl1b* or *lpl2b* genes were found on the other paralogon region.

The syntenic blocks around the *lpl1a* and *lpl1b* genes reveal the presence of diverse adjacent genes. Genes adjacent to *lpl1a* include *tpm4a*, *hsh2d* (upstream), and *adam10-like*, *safb*, *fam32a*, *mrpl54*, and *mal2* (downstream), which are conserved in all teleosts, except for Mexican tetra (*A. mexicanus*), which only retains downstream genes in synteny. However, the order of adjacent genes to *lpl1a* in Atlantic herring (*C. harengus*), Cod, Japanese medaka, and Nile tilapia differs from that in other species, showing variations in the arrangement of the *lpl1a* gene environment. Despite these particularities, the comparison of all orthologous regions shows that the syntenic organization of the gene group *tpm4-lpl1a-lpl2a-adam10-like* is highly conserved among the species analyzed.

The genes adjacent to *lpl1b*, *tpm4*, *rab8a*, *cib3* (upstream), and *ap1m1*, *klf2b*, *eps15l1*, *crt3*, *rx2* (downstream) are conserved in all teleosts that retained this copy, except for Mexican tetra, which shares only upstream genes in synteny. However, despite these variations, the syntenic organization of *tpm4*-*lpl1b*-*ap1m1* is highly conserved among the teleost species analyzed.

Fishes in the *Euteleostei* clade generally have genomic regions with a specific set of genes (*rab8a*, *tax1bp3*, *pole4*, *nim1k-like*, and *ccl25b*) adjacent to *lpl1a*, and to *lpl1b* (*admp*, *pnhd*, *nim1k-like*, and *ccl25b*), which is shared with the Early branching teleost *P. kingsleyae*, but is not found in the *lpl1a* and *lpl1b* genomic regions of Otocephala fishes. Overall, fishes from the Otocephala clade share fewer *lpl1a* and *lpl1b* neighboring genes (5 to 10 and 5 to 12, respectively) in synteny with the spotted gar reference genomic region, compared to fishes from the Euteleost clade (13 to 24 and 10 to 12, respectively).

Even in closely related species, such as Characiform representatives, the *lpl1a* and *lpl1b* genomic regions show notable variations in neighboring genes. The *lpl1a* genomic region of tambaqui (*C. macropomum*) shares 10 genes with the spotted gar reference, compared to five genes in the Red-bellied Piranha (*P. nattereri*) and eight genes in Mexican tetra. When using the Tambaqui *lpl1a* genomic region as a reference, variations among Otocephala fishes become more evident. The order of 13 out of 15 upstream genes is perfectly conserved between tambaqui and the Red-bellied Piranha *lpl1a*. Some of these genes are in synteny with more distantly related species, such as the Electric eel (*E. electricus*), Channel catfish (*I. punctatus*), and Zebrafish (*D. rerio*), but none of these genes are present in the Mexican tetra and Atlantic herring *lpl1a* regions. In contrast, tambaqui *lpl1a* shares only 3 downstream genes in synteny with the Red-bellied Piranha, while the order of 10 downstream genes (out of 15) is strongly conserved compared to the Channel catfish, Electric eel, and Zebrafish. Unlike this scenario, the order of *lpl1a* neighboring genes in euteleost fishes, such as Cod, Japanese medaka, and Nile tilapia, shows more conservation of both upstream and downstream gene blocks.

The neighboring genes of tambaqui *lpl1b* are in strong synteny with the Red-bellied Piranha, Channel catfish, and Electric eel in both upstream and downstream regions, despite some small variations. However, Mexican tetra *lpl1b* does not share any downstream neighboring genes with tambaqui *lpl1b*, while Zebrafish and euteleosts with *lpl1b*, such as Cod and Northern pike, share only three upstream genes in synteny with tambaqui *lpl1b*.

### 3.4. Characterization of Tambaqui Lpl1a, Lpl1b, and Lpl2a Protein Sequences

Several key amino acid residues, previously observed for vertebrate LPL, were identified in tambaqui Lpl1a, Lpl1b and Lpl2a protein sequences based on the alignment with human LPL (Figure 4, sequence numbers refer to human LPL numbers). The three tambaqui lipoprotein lipases have a larger amino-terminal domain (residues 1–340), the region responsible for catalysis, and a smaller carboxy-terminal end (residues 341–476) required for binding to the lipoprotein substrate. The conserved active site triad (Ser-159, Asp-183, and His-268), the oxyanion hole (Trp-82, Leu-160), a heparin binding domain (RKNR) and a N-linked glycosylation (Asn-X-Ser/Thr, where X can be any amino acid) located at Asn-386, are highly conserved among human LPL and tambaqui Lpls. However, the glycosylation site at Asn70 was shared only between human LPL and tambaqui Lpl2a, while unique or divergent potential glycosylation sites were predicted in tambaqui Lpl1a (two sites), Lpl1b (five sites). Additional heparin binding sites were identified in tambaqui Lpl1a, Lpl1b, and Lpl2a sequences sharing 75%, 66%, and 55% of identity.

Some important structure domains of tambaqui Lpls, such as the polypeptide lid region (residues 245–265), and the tryptophan-rich lipid-binding region (residues 412–422), share less than 50% of identity with human LPL. The alignment showed that tambaqui Lpl1a share four tryptophan residues with human LPL (Trp409, 417, 420, and 421), while Lpl2a shares three Trp residues with a substitution of Trp417 by a Lysine, and Lpl1b shares only one Trp residue with a replacement of Trp417 by a serine and two crucial tryptophan residues (Trp420 and Trp421) that correspond to Phenylalanine residues. The alignment of the polypeptide lid region of tambaqui Lpl1a, Lpl1b, and Lpl2a, which consist of 21 amino acids with no gaps, showed 33.33%, 47.62%, and 28.57% of identity with human LPL, respectively.

Both human LPL and tambaqui Lpl2a possess ten cysteine residues involved in the formation of five disulfide bridges (Cys-54 and Cys-67, Cys-243 and Cys-266, Cys-291 and Cys-302, Cys-310 and Cys-445, Cys-462 and Cys-466) and a Pro285 residue. In contrast, tambaqui Lpl1a and Lpl1b lack the disulfide bridge (two Cys residues) in the C-terminal domain and the Pro285 residue corresponds to a Gly residue.

We detected variations in the physicochemical properties among tambaqui Lpls (Appendix A). The Lpl1a variant has a molecular weight of 57,713.03 Da and a pI of 8.23. The instability index is 40.82, classifying this variant as unstable. The Lpl1b variant has a molecular weight of 60,016.45 Da and a pI of 8.22, with an instability index of 32.13, classifying it as stable. The Lpl2a variant has a molecular weight of 56,522.42 Da and a pI of 8.51, with an instability index of 32.36, classifying it as stable.

### 3.5. Comparisons of Lpls with Respective Orthologous

An expanded analysis was performed by aligning Lpl1a, Lpl1b, and Lpl2a from tambaqui with 32 vertebrate sequences (Appendix A). The catalytic triad (Ser-159, Asp-183 and His-268) is conserved across all Lpls analyzed, including LPLs, LPL2 and teleost WGD paralogues Lpl1a, Lpl1b, and Lpl2a, except for Lpl2a from *P. hypophthalmus* which has a substitution of His-268 to Gln residue, and *E. electricus* in which Ser-159 corresponds to Asn residue. The difference in cysteine residues between Lpl1 3R paralogues (8 Cys residues) and Lpl2a (10 Cys residues) is conserved across teleosts. However, LPL1 and LPL2 from *C. milii* share ten cysteine residues, a feature conserved in LPL from mammals and in Lpl2a from teleost fishes. Moreover, tambaqui Lpl2a contains the pro285 residue, together with *A. mexicanus* and *P. hypophthalmus*, which is an exception since this feature is conserved in both LPL1 and LPL2 from primitive representatives such as Elephant shark (*C. milii*) but absent in most teleost and non-teleost fishes.

An insertion of three amino acids was detected in the polypeptide lid region of Lpl1a from euteleosts (Figure 5) such as *G. morhua*, *O. niloticus*, *M. murdjan*, *P. fluviatilis*, *G. aculeatus*, *D. labrax*, *T. rubripes*, *Scophthalmus maximus*, *Cynoglossus semilaevis*, and *O. latipes*. The N-terminal residues (Ala194, Arg 197, Ser199, Asp201, and Asp202) are highly conserved across primitive LPL1 and LPL2 from *C. milii*, LPL from mammals, LPL2 from non-teleosts, and in Lpl2a from teleosts, with the exception of *P. hypophthalmus* where Arg197 corresponds to Val residue. In contrast, teleost Lpl1a maintained Ser199, Asp201, and Asp202 with substitutions at residues Ala194 and Arg197, while in Lpl1b the four residues are highly conserved except for substitutions at residues Ala194.

Analysis of the physicochemical properties of LPLs (Appendix A) from different species reveals the conservation and variation in these enzymes across vertebrates and teleost fish. LPLs (LPL1/LPL2) vary in molecular weight between species, generally ranging from 53,000 to 60,000 Da, indicating relative conservation of LPL protein size across vertebrates and teleost fish. Most Lpl1 have pIs between 8.22 and 8.80, indicating a slightly basic nature, with some exceptions. All tambaqui Lpls follow this trend with pIs between 8.22 and 8.51. In contrast, most Lpl2 have pIs between 6.78 and 7.95.

Instability indexes (II) varied between LPL variants in different species. In teleosts, at least one copy of Lpl1 is unstable and one is stable, except for *C. harengus* and *E. electricus*, which both have stable copies. Lpl1a is generally unstable in fish with two copies of Lpl1 (e.g., tambaqui and *G. morhua*), but can be stable in some cases (e.g., *M*. *murdjan*). Lpl1b is generally stable in fish and Lpl2a preserves stability in all teleost fish analyzed. Comparison with LPLs from primitive species indicates that instability is not exclusive to a single evolutionary lineage, appearing in both more primitive vertebrates (such as Coelacanth) and humans, as well as in non-teleost fishes such as *E. calabaricus* and *L. oculatus*.

### 3.6. Comparative Analysis of Glycosylation Sites Across Vertebrate LPLs

Although these are two ancient paralogous genes that share similarities in glycosylation sites retained throughout vertebrate evolution, our comparative analysis identified two distinct maps for LPL1 Table 1 and LPL2 Table 2, with the Elephant Shark as the most primitive reference. The complete data for these maps can be accessed via Mendeley Data https://data.mendeley.com/datasets/n9ys3nwy4k/1 (published on 10 March 2025).

The comparative analysis of potential N-glycosylation sites for vertebrate LPL1 and LPL2, including 3R paralogues (Lpl1a, Lpl1b, Lpl2a, and Lpl2b), has shown overall 26 and 15 sites, respectively. The LPL1 sequence from *C. milii*, contains five N-glycosylation sites (designated as sites 6, 13, 16, 18, and 19); however, only site 19 has been predominantly retained for all vertebrate LPL/LPL1 and LPL2 (and respective 3R paralogues) sequences examined.

While *C. milii* LPL1 (Table 1) shared glycosylation sites with other primitive species such as *L. chalumnae* (sites 13, 17, and 19), *E. calabaricus* (sites 13 and 19) and *L. oculatus* (sites 16, 18, and 19), the presence of additional unique or divergent sites (i.e., 1 and 15) indicates a species-specific glycosylation profile. Additionally, N-glycosylation site 8 was restricted to *L. chalumnae* and *L. oculatus* LPL1, and conserved in teleost Lpl1a (with an exception of *G. morhua* and *O. latipes*) and Lpl1b (with an exception of *E. electricus* and *P. hypophthalmus*).

The total number of sites found in 13 teleost fish Lpl1a sequences varied between 2 (*D. rerio*), 3 (*C. harengus*, *E. electricus*, *A. mexicanus*, *P. nattereri*, *C. macropomum*, *E. lucius*, *G. morhua*), 4 (*S. formosus*, *C. chanos*, *P. hypophthalmus*), and 5 (*O. niloticus*, *O. latipes*). Exclusively, the Lpl1a sequences from Otocephala fishes (except *C. harengus* and *D. rerio*) share the N-glycosylation site 7, while sites 4, 5, and 22 were found only in fishes from Euteleostei clade. Unique sites were found in Lpl1a sequences from *C. harengus* (site 23), *C. chanos* (site 14).

In contrast, Lpl1b retained two N-glycosylation sites, also found in *C. milii* LPL1, designated as site 6 which is highly conserved among the 9 teleost sequences examined, and site 16 which is restricted to some fish species except for *E. electricus*, *P. hypophthalmus*, *P. nattereri*, and *G. morhua.* Moreover, the N-glycosylation site 11 was detected exclusively in Lpl1b from Otocephala fishes, with an exception for *C. harengus* and *A. mexicanus*. Considering all glycosylation sites detected in teleost fish Lpl1b sequences, the total number of sites varied between four (*P. hypophthalmus*), five (*C. chanos*, *E. electricus* and *P. nattereri* and *G. morhua*), six (*C. harengus*, *A. mexicanus* and *C. macropomum*), and eight (*D. rerio*).

The LPL2 (Table 2) from *C. milii* contains six N-glycosylation sites, although only two of these sites (designated as sites 7 and 11) have been predominantly retained for LPL2 from *L. chalumnae*, non-teleosts and teleosts Lpl2a. The N-glycosylation site 7 was conserved in almost all Lpl2a sequences analyzed, except for *E. lucius*. In contrast, the site N-glycosylation site 9 was less conserved being absent in *L. chalumnae*, *L. oculatus*, *D. rerio*, *A. mexicanus*, *P. nattereri*, *C. macropomum* and *E. lucius*. Both sites 7 and 9, detected in Lpl2a, correspond to sites 13 and 16, also found in some LPL1 and Lpl1b sequences. As mentioned above, the N-glycosylation site 9 was highly conserved in all vertebrate LPL2, including teleosts Lpl2a sequences examined, which corresponds to site 19 also found in all vertebrate LPL/LPL1 sequences. Considering all glycosylation sites detected in teleost fish Lpl2a sequences, the total number of sites varied between 2 (*C. macropomum*, *P. nattereri* and *E. lucius*), 3 (*S. formosus*, *C. harengus*, *P. hypophthalmus*), 4 (*C. chanos*, *D. rerio*, *G. morhua* and *O. latipes*), 5 (*E. electricus*, *A. mexicans*), and 6 (*O. niloticus*).

### 3.7. Tissue Distribution of Tambaqui lpl1a, lpl1b and lpl2a mRNA

The relative expression levels of tambaqui *lpl1a*, *lpl1b*, and *lpl2a* mRNA were quantified by RT-qPCR in 12 tissues, including the liver, brain, gonads, stomach (anterior, middle, and posterior), intestines (anterior, middle, and posterior), pyloric caeca, muscle, and heart (Figure 6). The results showed that *lpl1a* was expressed in all analyzed tissues, although at different levels, with the highest expression in the liver, followed by the anterior intestine, muscle, and heart, respectively. In contrast to *lpl1a*, the expression level of tambaqui *lpl1b* was highest in the gonads, followed by the anterior, posterior, and middle intestines. No *lpl1b* transcripts were detected in the liver, while in the muscle and anterior and posterior stomachs, *lpl1b* was undetected in some individuals. *lpl2a* was expressed in all tissues analyzed, similar to *lpl1a*, with the highest levels observed in the liver, followed by the muscle and anterior and posterior intestines.

### 3.8. Comparison of lpl1a, lpl1b and lpl2a mRNA Tissue Distribution Between Non-Teleost and Teleost Fishes

Our comparative analysis showed that there is no conserved expression pattern of the *lpl1a*, *lpl1b*, and *lpl2a* genes among fish species, and the tissues involved in lipid metabolism vary between species (Figure 7). In *L. oculatus*, a non-teleost representative, *lpl1* and *lpl2* are predominantly expressed in the heart, bones, brain, and gonads. In teleosts, *lpl1a* was more abundant in the liver of most species analyzed (*D. rerio*, *A. mexicanus* surface fish, *P. altivelis*, *G. morhua*, *O. latipes* and *P. fluviatilis*). However, in *O. bicirrhosum*, *A. alosa*, *P. hypophthalmus,* and *A. mexicanus* (cave fish) *lpl1a* reaches the highest levels in the testis, brain, bones and ovary, respectively, with the lowest levels in the liver.

With exception of *D. rerio*, in which *lpl1b* was more abundant in gonads (testis) and undetectable in the liver, the *lpl1b* reached the highest levels in the same tissues as *lpl1a*. A greater variation in the gene expression pattern of *lpl2a* was observed among different fish species, being more abundant in testis of *O. bicirrhosum* and *P. hypophthalmus*, heart of *A. alosa*, *O. latipes* and *P. fluviatilis*, intestine of *D. rerio*, ovary of *A. mexicanus* from cave, liver of *A. mexicanus* from surface and *E. lucius*, bones of *P. altivelis* and gills of *G. morhua*.

We detected several distinct expression patterns of *lpl1a*, *lpl1b*, and *lpl2a* in teleost fish tissues: (1) The three genes have differential expression between tissues, for example, in *D. rerio*, *lpl1a* is more expressed in the liver, *lpl1b* is more expressed in the testis and *lpl2a* is more expressed in the intestine. (2) *lpl1a* and *lpl1b* are more expressed in the same tissue, but *lpl2a* is more expressed in another, for example, in *A. alosa*, *lpl1a* and *lpl1b* are more expressed in the brain and *lpl2a* is more expressed in the heart; in *P. hypophthalmus*, *lpl1a*, and *lpl1b* are more expressed in the bones and *lpl2a* is more expressed in the testis; in *G. morhua*, *lpl1a*, and *lpl1b* are more expressed in the liver and *lpl2a* is more expressed in the gills. Additionally, in species that have two copies in the genome, such as *O. latipes* and *P. fluvialitis*, *lpl1a* is more expressed in the liver and *lpl2a* is more expressed in the heart. (3) *lpl1a* and *lpl2a* are more expressed in the same tissue, and *lpl1b* is more expressed in another. For example, in tambaqui *lpl1a* and *lpl2a* are more expressed in the liver, and *lpl1b* is more expressed in gonads. This pattern is unique to tambaqui among the species that have all three copies. (4) All existing copies in the species’ genome are more expressed in the same tissue; for example, in *A. mexicanus* surface fish and *A. mexicanus* cave fish, where *lpl1a*, *lpl1b*, and *lpl2a* are more expressed in the gonads and liver, respectively. In fish with two copies, such as *O. bicirrhosum*, *lpl1a*, and *lpl2a* are more expressed in the testis.

## 4. Discussion

Our study employs a comprehensive approach that combines phylogenomic, syntenic, and gene expression analyses to confirm multiple lipoprotein lipase gene (*lpl*) copies in the tambaqui genome, elucidating their origin and relationship to teleost-specific duplication events. These genes are organized into ten exons and produce proteins with conserved structural domains, indicating functional conservation. This finding advances the understanding of *lpl* gene evolution within the Actinopterygii class, which previously focused on lower gene copy numbers in other fish species [21,45].

### 4.1. Evolution and Phylogeny of LPL Genes

The phylogenomic analysis positioned the ‘LPL-like’ proteins as a distinct member of the well-known LPL clade, a finding that builds on the work previously reported by [2]. Our results support that these two genes originated from an ancient tandem duplication event in the last common ancestor of cartilaginous and bony fishes, as indicated by their conserved synteny in the genomes of sharks and coelacanths. To facilitate discussion and distinguish between these two paralogs, we refer to the copy orthologous to mammalian LPL as LPL1, and to its tandemly duplicated counterpart as LPL2 throughout the manuscript. Following the ZFIN guidelines, we recognize that gene symbols should not be assigned arbitrarily; thus, we emphasize that this nomenclature is used solely to distinguish between the products of an ancient tandem duplication retained in several fish lineages. However, prior studies in medaka, pufferfish, and red seabream have adopted similar naming schemes to differentiate functionally and phylogenetically distinct copies [23,46,47].

Our phylogenetic analysis indicates that *LPL1* corresponds to the ortholog of mammalian *LPL*, while *LPL2* represents a now-lost paralog in tetrapods, retained in several aquatic vertebrates. One type of genomic change with potential to affect phenotypic evolution is the inactivation or loss of ancestral protein-coding genes [48]. In this case, the *LPL2* was likely lost in tetrapods, due to physiological and dietary changes during terrestrial evolution [49,50,51]. In contrast, similar to sharks and coelacanths, non-teleost Actinopterygii retained both *LPL1* and *LPL2*, possibly due to continuous lipid metabolism needs in aquatic environments [52,53,54,55] and the adaptive advantages provided by *LPL2*.

### 4.2. Gene Retention, Loss, and Functional Diversification

The teleost-specific duplication (3R) further diversified the *LPL1* into *lpl1a* and *lpl1b*. Interestingly, *lpl1a* was retained across all teleosts analyzed, whereas *lpl1b* was absent in *S. formosus* and in most euteleosts with exception of salmonids, *G. morhua* and *d*. This suggests that some Early branching teleost fish lost the *lpl1b* paralog after the diversification of Clupeocephala fish, while in euteleosts, this copy was lost before the diversification of neoteleosts such as *O. latipes* and *O. niloticus*.

However, the Otocephala group, including tambaqui, consistently retains *lpl1a*, *lpl1b*, and *lpl2a*. This retention underscores the functional significance of these genes for the metabolic versatility of the Otocephala group, suggesting that the persistence of these paralogs provides adaptive advantages in response to environmental or physiological demands [56]. Moreover, the diversification of 3R duplicated genes likely facilitated processes of subfunctionalization and neofunctionalization [34], a phenomenon that enhances the flexibility and adaptability of metabolic processes [57]. This mechanism allows the Otocephala group, especially neotropical fishes, to efficiently exploit diverse ecological niches [58,59,60], supporting the hypothesis that gene duplication contributes to functional diversification and metabolic innovation.

Furthermore, the *LPL2* gene also underwent duplication in teleosts, resulting in the formation of *lpl2a* and *lpl2b*. While *lpl2a* was widely retained across teleosts, *lpl2b* was found only in *P. kingsleyae*, an Early branching teleost species. However, its absence in other Early branching teleosts, such as *S. formosus*, *A. anguilla*, and *M. cyprinoides*, suggests that *lpl2b* was extensively lost in multiple early teleost lineages and in the common ancestor of Clupeocephala, shortly after the third round (3R) whole-genome duplication. This pattern of gene retention and loss follows broader trends in genome duplications, where non-functionalization leads to rapid gene loss after genome duplication [56]. Thus, the rapid loss of *lpl2b* in certain Early branching teleosts may represent a typical evolutionary trajectory, where nonfunctional duplicates are swiftly eliminated, contributing to the streamlining of the genome.

Synteny analysis revealed a high degree of conservation in the genomic regions surrounding the *LPL1* and *LPL2* genes between Spotted gar and other aquatic vertebrates, except tetrapods, supporting the hypothesis of an ancestral genomic configuration. This conservation reinforces the inferred evolutionary history of the *lpl1* and *lpl2* genes and highlights the relevance of duplication origin and conserved positional context in understanding the evolution of these gene lineages.

The 3R duplication resulting in *lpl1a–lpl2a* and *lpl1b–lpl2b*, located in different chromosomes, is evident from the conserved syntenic blocks observed in both Early branching and derived teleosts. Variations in adjacent genes, particularly the adaptive rearrangements seen in different teleost lineages such as Ostariophysi and Euteleosts *lpl1a*, reflect ongoing evolutionary pressures shaping these regions. Changes in the genomic neighborhood after the split from the common ancestor are linked to divergences in gene expression levels in different tissues, potentially leading to phenotypic divergences [61]. The specific retention of adjacent genes in the *lpl1a* and *lpl1b* regions of tambaqui (*C. macropomum*) and red-bellied piranha (*P. nattereri*), both members of the Serrasalmidae family, compared to other teleosts, may reflect conserved genomic architecture, which is more likely to be maintained between closely related species. These patterns may reflect structural constraints or lineage-specific regulatory evolution, rather than direct functional adaptation.

### 4.3. Structural and Functional Insights of Lpl Proteins

The study of tambaqui Lpl1a, Lpl1b, and Lpl2a protein sequences reveals crucial insights into their functional roles and evolutionary adaptations. The conserved catalytic triad (Ser-159, Asp-183, and His-268), and oxyanion hole (Trp-82 and Leu-160) ensure proper catalytic function, while heparin-binding motifs are essential for lipid metabolism and enzyme anchoring to the endothelial surface. These features are shared across vertebrate LPLs, indicating a common mechanism of action [1,2,21,62].

Comparative analysis revealed that in primitive fish species, such as *C. milii* and *L. chalumnae*, both LPL1 and LPL2 retained ten cysteines, suggesting a conserved ancestral state critical for LPL structure and function. The divergence in cysteine numbers between LPL1 and LPL2 begins in non-teleost fish and is preserved after 3R in teleosts. Surprisingly, Lpl2a contains ten cysteine residues (forming five disulfide bridges), contrasting with the eight found in Lpl1a and Lpl1b. The C-terminal disulfide bond and Pro285 have been regarded as critical for lipase stability at 37 °C in mammals [22], but it has been demonstrated that these features are not present in fish Lpl1 [62].

However, this structural difference, along with the presence of Pro285 in Lpl2a from tambaqui and other few teleosts could enhance enzyme stability and functionality under varying environmental conditions, which is consistent with our findings on the physicochemical properties of these enzymes. LPL stability is maintained in Lpl2a, whereas Lpl1a tends to be unstable and Lpl1b stable in most fish with two Lpl1 copies. In species with only one Lpl1 copy, this copy tends to be stable, ensuring the protein’s essential functionality. The presence of two Lpl1 copies may allow functional specialization, with one copy remaining stable and the other exploring evolutionary variation [56]. While the instability index is a theoretical prediction and does not directly determine in vivo protein degradation, lower stability may suggest a higher turnover rate, potentially affecting enzyme availability and activity under different physiological demands [63]. These variations may reflect specific adaptations in the optimal activity of lipases and temperature stability, tailored to each species physiological and environmental needs [64,65,66,67]. Additionally, such variation might also be tissue-dependent, as certain tissues could require more dynamic regulation of lipid metabolism than others [18].

Conserved Trp residues in LPL, LPL2, and teleost Lpl1a indicate their critical role in protein function, such as substrate recognition [68]. Variations in Trp residues, particularly in tambaqui and other fish Lpl1b, suggest functional divergence. Moreover, the replacement of Trp409 in Lpl2a sequences from tambaqui and other ostariophysians reflects evolutionary adaptation to specific environments or diets, influencing lipid-binding properties and lipolysis efficiency.

For example, the feeding ecology of species like the tambaqui (*C. macropomum*) and pacu (*Piaractus mesopotamicus*), both of which belong to the family Serrasalmidae, exhibit dietary flexibility—frugivory in the wet season and detritivore during the dry season in tambaqui [69,70], and a primarily herbivorous diet with occasional consumption of invertebrates in pacu [71]. These dietary shifts likely drive adaptations in lipase activity and may influence the evolution of lipases such as Lpl1b, optimizing their ability to adapt to varying metabolic demands, environmental conditions, and diet compositions, enhancing lipid digestion and energy utilization in these species.

The polypeptide lid region of tambaqui Lpl1a and Lpl1b shares 72% identity, indicating a close evolutionary relationship and similar functions. In contrast, tambaqui Lpl2a has only 31.82% and 27.27% identity with Lpl1b and Lpl1a, respectively, suggesting greater divergence and potentially specialized functions. The lid protects the active site and is responsible for catalytic activity, with differences in lid sequence leading to significant changes in substrate selectivity, activity, and thermostability [72]. Interestingly, the lid region’s conservation in Lpl1a among Ostariophysi fish highlights its crucial role in enzymatic activity and lipid substrate interaction. In contrast, euteleostean fish exhibit a 3-amino-acid insertion in the lid region, reflecting specific adaptations of these groups to their environments or diets.

Differences in the lid region between mammalian PL (pancreatic lipase) and fish PL have been previously reported. In most fish species, Ile was replaced by Ser, Lys, or Arg in the lid domain of pancreatic lipase, which might also influence its lipolytic ability [57,73]. Moreover, similar to what was found for LPLs, the amino acid corresponding to Leu in mammalian PL was replaced by Ile or Ala in some fish species, such as spotted gar, European eel, catfish, and northern pike, but was absent in some euteleost fish species (e.g., Atlantic cod, mandarin fish, seabass, pufferfish, and Japanese flounder) [57].

The lid regions of Lpl1b and Lpl2a are more variable within and between teleost lineages, with *G. morhua* and *E. lucius* showing the most divergent sequences. In *Cebidichthys violaceus*, extensive genetic variation and adaptive amino acid variation in amylase and carboxyl ester lipase suggest multiple mechanisms underlying the novel derived dietary physiology [74], supporting the notion of adaptive modulation and nutrient balancing in response to dietary needs. This further underscores how variations in the lid region of lipases are linked to specific environmental or dietary shifts in various teleost lineages.

Glycosylation patterns in LPL and LPL2 variants exhibit a mix of conservation and functional diversification. The conserved glycosylation site at Asn-386 in tambaqui Lpl1a, Lpl1b, and Lpl2a suggests important roles in protein stability, secretion, and catalytic activity, consistent with previous predictions of this site in LPLs from other vertebrates [75]. The presence of multiple glycosylation sites in tambaqui Lpl1b, compared to fewer in Lpl1a, indicates differential post-translational modification requirements. Variations in glycosylation sites across teleost Lpl1a and Lpl1b orthologs point to adaptive divergence for specific functions in different tissues, environments, and ecological niches occupied distinctively by Otocephala and Euteleosts. Similarly, although teleost fishes Lpl2a shares three main glycosylation sites, additional unique sites and variations (i.e., the loss of conserved site 7 in tambaqui Lpl2a) have led to unique glycosylation patterns among teleost orthologs indicating specie-specific adaptations.

### 4.4. Tissue-Specific Expression and Ecological Adaptations

Gene expression studies in fish have focused mainly on the *lpl1a* gene, also referred to as “*lpl*” or “*lpla*”, which has been detected in various tissues of marine and freshwater fish but with different expression patterns. Apart from adipose tissue, *lpl* has the highest expression in the liver of adult *D. rerio* [62], *O. latipes* [46], and *S. chuatsi* [76], in the muscle of *A. dobryanus* [21] and *O. clarki* [32], in the stomach of *Coilia nasus* [20], and in gonads of *D. labrax* [77]. To date, there are no reports on the gene expression of *lpl1b*, but one study reported the occurrence of a second type of *lpl* in *O. clarki* (*ctlpl2*), referred to here as *lpl2a*, with predominant expression in the granulosa cells of ovarian follicles [32]. Additional studies have also identified a second *lpl-like* gene in other teleosts, including medaka (*O. latipes*) [46], red seabream (*P. major*) [23], and pufferfish (*T. rubripes*) [47], further supporting the presence and retention of *lpl2a* in multiple fish lineages. In these studies, the *lpl1a* and *lpl2a* paralogs were referred to as *lpl1* and *lpl2*, and *lpl2* showed expression patterns similar to those of *lpl1*, suggesting possible functional overlap or coordinated regulation between the two genes.

The presence of *lpl1a*, *lpl1b*, and *lpl2a* transcripts in tambaqui tissues, such as the liver, brain, gonad, stomach, intestine, pyloric caeca, heart, and muscle, suggests overlapping or complementary roles in regulating lipid metabolism. The differential expression patterns of *lpl1a*, *lpl1b*, and *lpl2a* mRNA levels in tambaqui tissues underscore the complexity and specialization of lipid metabolism across various organs. In tambaqui, *lpl1a* and *lpl2a* transcripts were more abundant in the liver, while *lpl1b* was predominantly detected in the gonads. This pattern is unique among the fish studied, indicating specific adaptations to the metabolic and reproductive demands of the tambaqui. These adaptations are especially critical for Neotropical species, especially for species with a migratory lifestyle such as the tambaqui, which face dynamic and varied environments [36,70,78].

The liver is crucial for lipid metabolism [79], and the high expression of *lpl1a* and *lpl2a* suggests a specialization in the hydrolysis of triglycerides to free fatty acids and glycerol. This allows efficient use of stored energy, essential for the growth and maintenance of health. The tambaqui, native to the rivers and lakes of the Amazon, is exposed to fluctuations in food availability [37,70] and the hepatic expression of *lpl1a* and *lpl2a* may represent an adaptive mechanism to optimize energy storage efficiency during periods of abundance and facilitate its mobilization during times of scarcity.

The expression of *lpl1b* in the gonads suggests a crucial role in the provision of lipids necessary for gamete development and maturation. This is in line with previous findings in other fish species, such as the sea bass, where *lpl* (cDNA from the ovary) has been shown to be highly expressed in the ovary, particularly when lipid reserves are being accumulated in the oocytes [80]. The prioritization of reproduction may be an evolutionary response to the environmental conditions of the Amazon, where energy must be directed efficiently to guarantee the continuity of the species [36].

Furthermore, these tissue-expression differences between *lpl1a* and *lpl1b* are consistent with the variations in the glycosylation patterns predicted for the tambaqui amino acid sequence, which may suggest tissue-specific adaptations. The higher number of predicted glycosylation sites for Lpl1b, compared to Lpl1a, could imply differentiated post-translational modifications, possibly influenced by the distinct physiological needs of the tissues. The preferential expression of *lpl1b* in the gonads, in line with the glycosylation patterns, suggests that glycosylations might contribute to modulating enzymatic activity, potentially adjusting the function of *lpl1b* in different physiological environments, such as gonadal development.

Our study revealed the differential expression of *lpl1a*, *lpl1b*, and *lpl2a* throughout the tambaqui gastrointestinal tract, including the stomach (anterior, middle, and posterior), intestine (anterior, middle, and posterior), and pyloric caeca. Surprisingly, the intestine exhibited the second highest abundance of *lpl1a*, *lpl1b*, and *lpl2a* transcripts, a unique pattern not observed in other fish species, based on our comparative analysis. This prominent intestinal expression suggests a crucial role of Lpls in the absorption and processing of dietary lipids in tambaqui. As an omnivorous species, the tambaqui consumes a wide range of food items, including fruits, seeds, and zooplankton [37,69,70], which can vary in lipid content. Therefore, the overlapping expression of *lpl1a*, *lpl1b*, and *lpl2a* in the intestine may allow tambaqui to efficiently adapt to these variations in diet composition, maximizing energy extraction from dietary lipids.

In line with the findings from the phylogenomic study and protein sequence analysis, comparisons with *lpl* paralogous copies (*lpl1a*, *lpl1b*, and *lpl2a*) in other fish species highlighted a large variation in tissue distribution patterns, reflecting specific adaptations in lipid metabolism across different species. Furthermore, the differential expression of paralogous copies in different tissues suggests subfunctionalization and partitioning of functions after gene duplication [34]. Interestingly, in some fish species, the liver is not the primary tissue for the expression of any of the *lpl* copies. In these cases, other tissues, such as the brain, bones, heart, and gonads, become the main target tissues (e.g., *L. oculatus*, *O. bicirrhosum*, *A. alosa*).

The distinct patterns of *lpl1a*, *lpl1b*, and *lpl2a* gene expression in surface (more expressed in the liver) and cave (more expressed in the ovary) populations of *A. mexicanus* provide a fascinating insight into the adaptive mechanisms that have evolved in response to different environmental pressures. In surface-dwelling fish, liver expression supports high metabolic activity and energy demands, while in cave-dwelling fish, ovary expression ensures reproductive success in nutrient-limited conditions. These differences highlight the remarkable ability of species to modify metabolic pathways and gene expression to thrive under varying environmental pressures.

Thus, the variability in the distribution of *lpl* copies and the adaptive strategies of different species are addressed in a way that complements the previously provided information about the evolution and diversification of *lpl* gene structures. These differences highlight the remarkable ability of fish to adjust their lipid metabolism in response to different environmental pressures and ecological demands, contributing to their diversity and evolutionary success across a wide range of aquatic ecosystems.

### 4.5. Implications for Aquaculture, Conservation, and Future Research

These findings also have significant implications for aquaculture and the conservation of cultured fish, especially Neotropical fish species such as tambaqui. Understanding the functional diversification of *lpl* genes and its copies, and their tissue-specific expression patterns can guide the development of targeted feeding strategies and breeding programs. For example, optimizing diets based on the lipid metabolism requirements of different life stages or environmental conditions could improve growth performance and overall health in tambaqui aquaculture. The diversity and unique characteristics found in the tambaqui *lpl* genes, as well as in other teleosts, may offer potential for the development of genetically modified fish in aquaculture. Moreover, our findings can guide the design and modification of fish lipases for specific applications in industries such as pharmaceutical, food and biofuels, where catalytic efficiency and enzyme stability are crucial.

## 5. Conclusions

In conclusion, our study provides new insights into the evolution and function of *lpl* gene copies in the tambaqui genome, shedding light on the complex interplay between gene duplication, structural adaptations, and tissue-specific expression patterns. The integration of phylogenomic, syntenic, and gene expression data offers valuable insights into the metabolic adaptations of tambaqui and the role of *lpl* genes in shaping the remarkable adaptability of teleost fish to diverse environments. These findings lay the foundation for future research on lipid metabolism regulation in teleosts, including the relatively underexplored Neotropical fish species, and have the potential to guide practical applications in aquaculture and conservation efforts.

## Figures and Tables

**Figure 1 genes-16-00548-f001:**
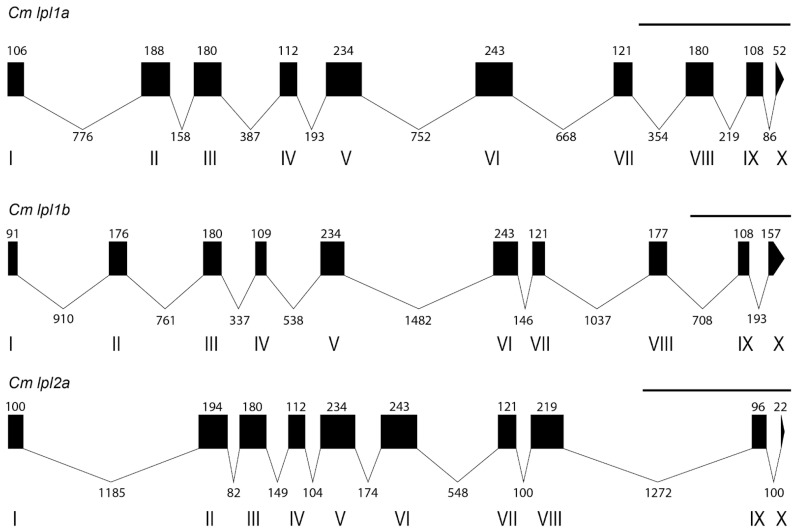
Genomic characterization of *C. macropomum lpl* paralogs. Schematic representation of the relative positions of introns and exons in *C. macropomum lpl1a*, *lpl1b*, and *lpl2a*. Exons are shown as boxed regions, while introns are represented by connecting lines. Exon positions are labeled with Roman numerals, and their lengths are indicated by cardinal numbers. A scale bar representing 1000 base pairs is shown above the diagram.

**Figure 2 genes-16-00548-f002:**
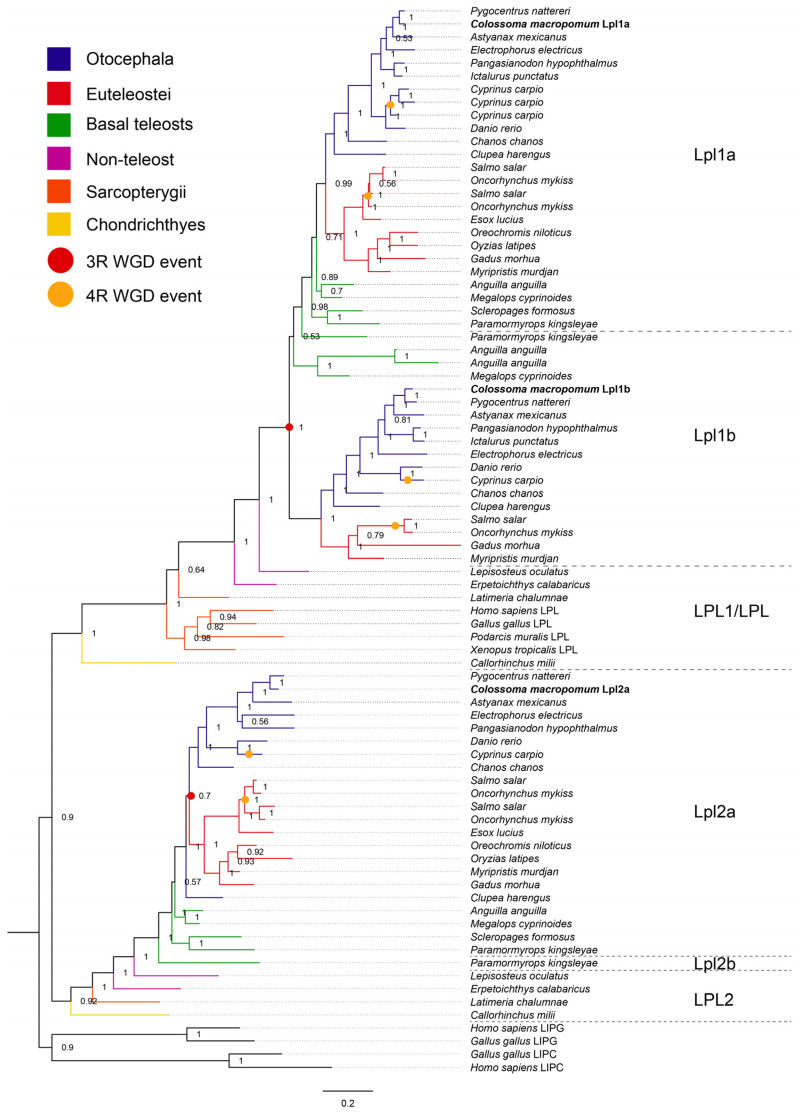
Phylogenetic tree of vertebrate lipoprotein lipase amino acid sequences. Phylogenetic tree constructed using Bayesian analysis based on 78 lipoprotein lipase sequences from 29 species representing chondrichthyans, sarcopterygians, and actinopterygians. Sarcopterygian LIPG and LIPC protein sequences were used as the outgroup. Node values represent Bayesian Inference (BI) posterior probabilities (below branches or right of slash). Whole-genome duplication events are indicated by dots. Red dots represent 3R events, which occurred after the first (1R) and second (2R) rounds of genome duplication, while orange dots represent 4R events, including the salmonid-specific (Ss4R) and carp-specific (Cs4R) duplications. The scale bar represents the average number of substitutions per site. Dotted lines connect taxon names to branch tips.

**Figure 3 genes-16-00548-f003:**
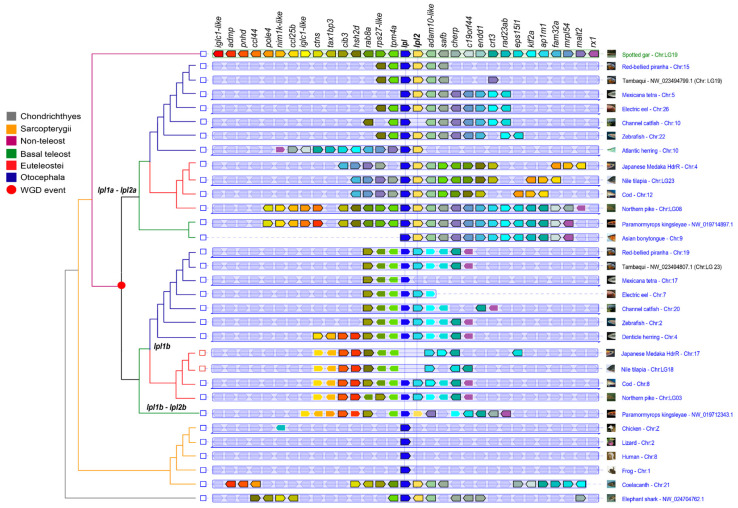
Synteny analysis of *lpl1a*, *lpl1b*, and *lpl2a* loci in *C. macropomum* (tambaqui) and representative vertebrates. Synteny analysis based on PhyloView in *Genomicus*, illustrating the genomic organization of *lpl1a*, *lpl1b*, and *lpl2a* loci in selected vertebrate species. The *LPL* and *LPL2* genes, both used as reference genes, are centered and aligned with a vertical black line. Orthologous genes across species are color-matched, with paralogs outlined in white, while their corresponding orthologs are outlined in black. Shaded genes represent non-orthologous genes relative to the spotted gar reference species (*L. oculatus*). A thin double-headed arrow beneath a gene block indicates that the gene order has been reversed compared to the ancestral orientation. On the left, a phylogenetic tree illustrates the evolutionary relationships of the reference gene, with red circle nodes marking duplication events of its ancestral version.

**Figure 4 genes-16-00548-f004:**
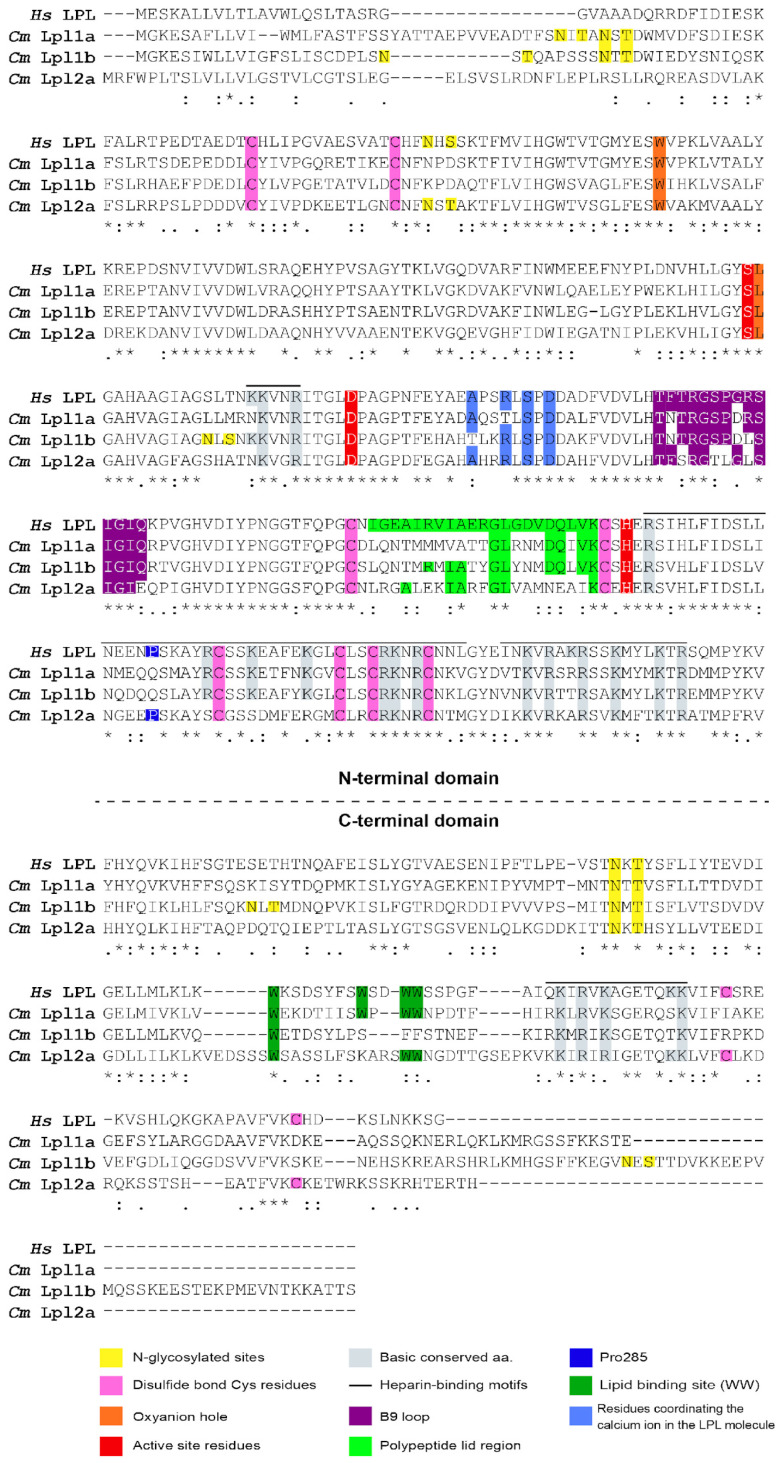
Multiple sequence alignment of Lpl1a, Lpl1b, and Lpl2a from *C. macropomum* with human LPL. Amino acid sequence alignment of tambaqui Lpl1a, Lpl1b, and Lpl2a using the human LPL sequence as a reference. Asterisks (*) indicate conserved residues among all sequences, while variable sites highlight potential functional or structural divergences among the paralogs. Key functional domains, including the catalytic triad and lipid-binding regions, are highlighted.

**Figure 5 genes-16-00548-f005:**
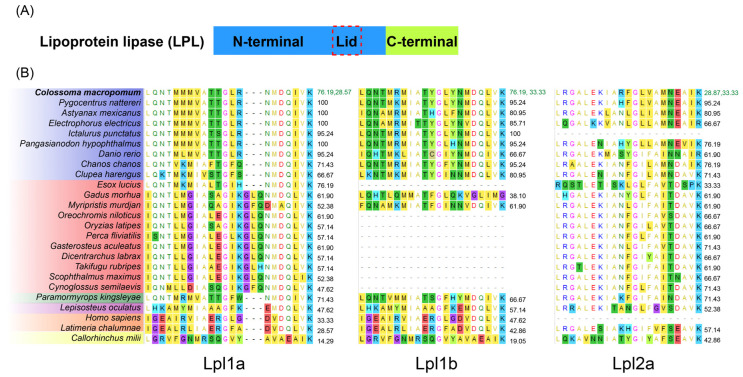
Structural and evolutionary comparison of the lid region in lipoprotein lipase proteins across vertebrates. (**A**) Schematic representation of the Lpl protein, highlighting the N-terminal and C-terminal domains, along with the lid region. (**B**) Amino acid sequence alignment of the lid region in tambaqui Lpl1a, Lpl1b, and Lpl2a with those of other vertebrates. Teleost groups are color-coded: Otocephala (blue), Euteleostei (red), and Early branching teleosts (green). Non-teleost actinopterygians are shown in purple, sarcopterygians in orange, and chondrichthyans in yellow. Percentage identities between tambaqui *Lpl* paralogs are indicated in green, while identities between each tambaqui *Lpl* and its respective orthologs are shown in black. Conserved sites are highlighted at the 50% threshold.

**Figure 6 genes-16-00548-f006:**
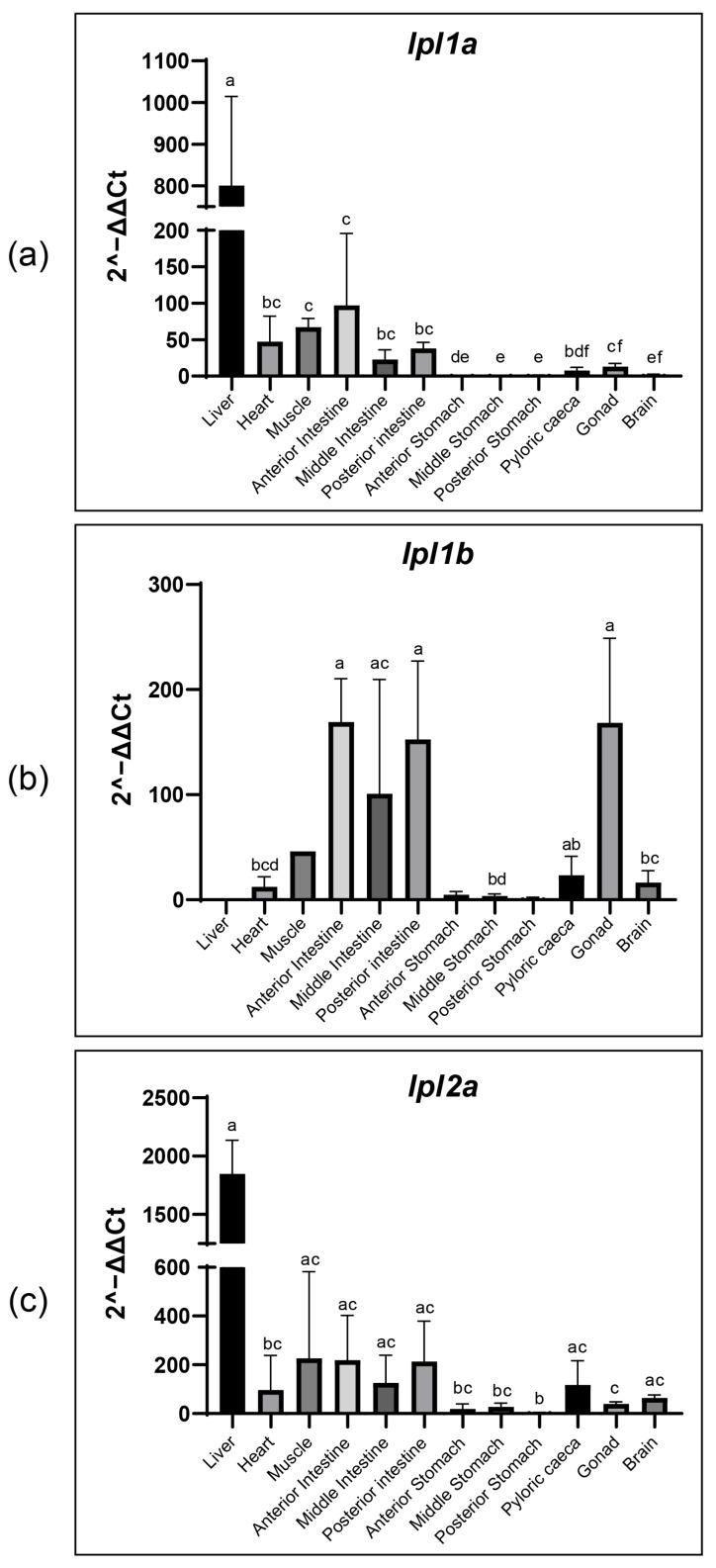
Tissue-specific expression of tambaqui *lpl* genes (*lpl1a*, *lpl1b*, and *lpl2a*) detected by qPCR. Expression of tambaqui *lpl1a* (**a**), *lpl1b* (**b**), and *lpl2a* (**c**) in different tissues including liver, heart, muscle, intestines, stomachs, pyloric caeca, gonads, and brain detected by qPCR. Relative expression data were calculated by the method 2^−ΔΔCt^. Results are represented as bar graphs. Variables with the same letter indicate no statistically significant differences between means. Variables with different letters are significantly different. The efficiencies obtained were 101.39% for *lpl1a*, 97.75% for *lpl1b*, 95.99% for *lpl2a*, and 100.88% for *β-actin*, based on five-point serial dilution curves.

**Figure 7 genes-16-00548-f007:**
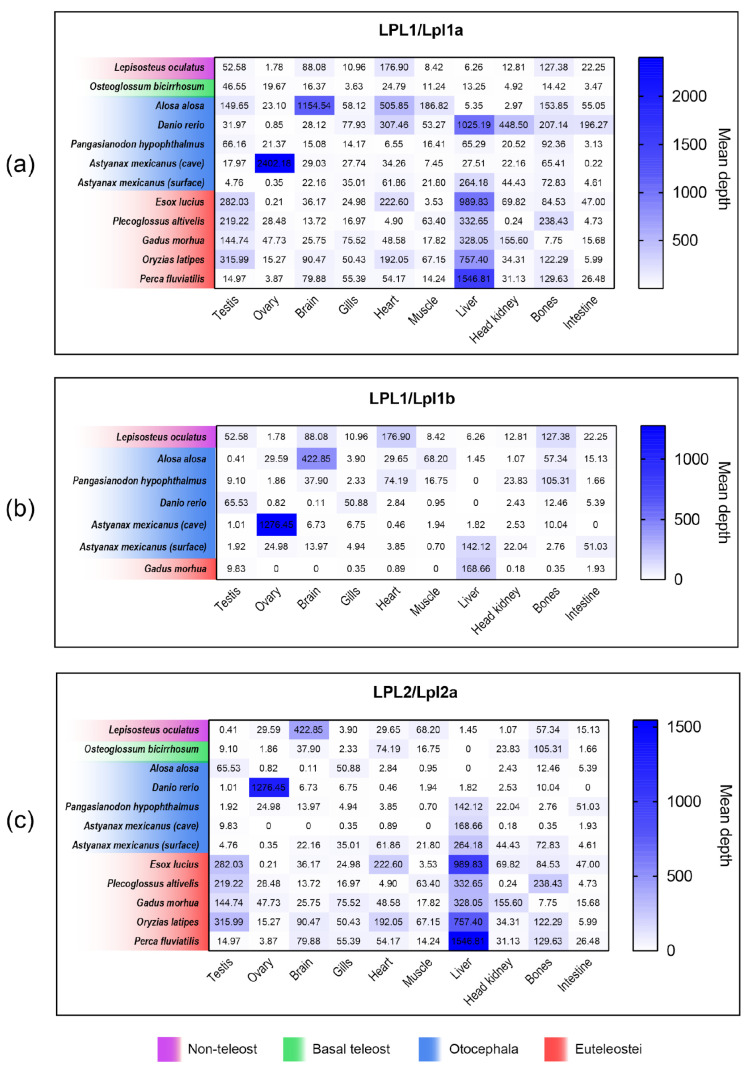
Heatmap of tissue-specific expression patterns of lipoprotein lipase genes in other fishes. Comparative analysis of tissue-specific expression patterns of *LPL1*/*lpl1a* (**a**), *LPL1*/*lpl1b* (**b**), and *LPL2*/*lpl2a* (**c**) in teleost and non-teleost fishes. The RNA-seq depth values were obtained from the PhyloFish database, providing insights into the expression profiles of these genes across different tissues in both teleost and non-teleost species.

**Table 1 genes-16-00548-t001:** Comparative glycosylation sites among vertebrate LPL, Lpl1a and Lpl1b.

Protein	Taxonomic Groups	Species	Site 1	Site 2	Site 3	Site 4	Site 5	Site 6	Site 7	Site 8	Site 9	Site 10	Site 11	Site 12	Site 13	Site 14	Site 15	Site 16	Site 17	Site 18	Site 19	Site 20	Site 21	Site 22	Site 23	Site 24	Site 25	Site 26	Site 27	Site 28	Total
LPL1	Chondrichthyes	*C. milii*						NTS							NIS				NRT		NET	NQT									5
	Sarcopterygii	*L. chalumnae*	NRT							NST					NLT				NWT			NKT									5
		*H. sapiens*													NHS							NKT									2
	Non-teleosts	*E. calabaricus*													NLT			NTS				NKT									3
		*L. oculatus*								NTT									NIS		NET	NKT									4
Lpl1a	Basal teleosts	*M. cyprinoides*								NTT												NTT					NGS				3
		*P. kingsleyae*						NST	NST							NKT						NAT			NES						5
		*S. formosus*			NTT					NTT												NTT									4
	Otocephala	*C. harengus*								NNT												NTT					NGS				3
		*C. chanos*							NST	NST							NKT					NTT									4
		*D. rerio*								NAT												NST									2
		*E. electricus*							NIT	NST												NTT									3
		*P. hypophthalmus*		NLS					NTT	NST												NIT									4
		*I. punctatus*							NIT	NST												NTT									3
		*A. mexicanus*							NIT	NST												NTT									3
		*P. nattereri*							NIT	NST												NTT									3
		*C. macropomum*							NIT	NST												NTT									3
	Euteleostei	*E. lucius*								NST												NST				NLS					3
		*G. morhua*			NTT		NST															NST									3
		*O. niloticus*				NTT	NET			NTT												NTT				NQS					5
		*Ö. Latipes*		NIS		NTT	NST															NNT				NIS					5
Lpl1b	Basal teleosts	*M. cyprinoides*								NST									NAT			NTT					NGS				4
		*P. kingsleyae*			NTT					NTT	NFS								NQT			NST									5
	Otocephala	*C. harengus*						NTT		NTT					NIS				NFS	NMS		NTT									6
		*C. chanos*						NTT		NST			NLT						NVS			NTT									5
		*D. rerio*						NST		NFT		NDS	NLT						NMT			NST							NQS	NTS	8
		*E. electricus*						NST			NHS		NLT	NVS								NTT									5
		*I. punctatus*						NST		NTT			NLS									NTT									4
		*P. hypophthalmus*						NST					NLT									NTT						NET			4
		*A. mexicanus*						NPT		NTT		NDS							NMT			NTT						NET			6
		*P. nattereri*						NST		NTT			NLS									NTT						NES			5
		*C. macropomum*						NST		NTT			NLS						NLT			NMT						NES			6
	Euteleostei	*E. lucius*																													
		*G. morhua*						NTT		NTT					NTT							NHT		NQT							5
		*O. niloticus*																													
		*Ö. latipes*																													

**Table 2 genes-16-00548-t002:** Comparative glycosylation sites among vertebrate LPL2, Lpl2a and Lpl2b.

Protein	Taxonomic Groups	Species	Site 1	Site 2	Site 3	Site 4	Site 5	Site 6	Site 7	Site 8	Site 9	Site 10	Site 11	Site 12	Site 13	Site 14	Site 15	Site 16	Site 17	Site 18	Site 19	Site 20	Total
LPL2	Chondrichthyes	*C. milii*			NTT						NAT	NVT		NHS		NKT					NQT		6
	Non-teleost	*L. chalumnae*									NTT					NRT							2
		*E. calabaricus*									NLT			NTS		NKT							3
		*L. oculatus*									NHT					NKT							2
Lpl2b	Basal teleosts	*P. kingsleyae*							NHT	NPT	NAT		NTS			NRT	NSS						6
Lpl2a	Basal teleosts	*A. anguilla*									NTT	NVT		NQS		NKT							4
		*M. cyprinoides*									NTT			NQS		NKT							3
		*P. kingsleyae*									NTT			NCS		NNT		NRS					4
		*S. formosus*									NAT			NHS		NRT							3
	Otocephala	*C. harengus*									NST			NQS		NRT							3
		*C. chanos*		NKT							NST			NRT		NKT							4
		*D. rerio*	NIT				NPS				NHT					NKT							4
		*E. electricus*						NLS			NTT			NQS	NGS	NKT							5
		*P. hypophthalmus*									NST			NQS		NTT							3
		*I. punctatus*																					
		*A. mexicanus*									NNT				NGS	NKT				NLS		NWS	5
		*P. nattereri*									NST					NKT							2
		*C. macropomum*									NST					NKT							2
	Euteleostei	*E. lucius*														NKT						NWS	2
		*G. morhua*		NAT							NST			NRS		NKT							4
		*O. niloticus*		NVT		NAT					NST			NSS		NKT			NVT				6
		*Ö. Latipes*		NST							NRT			NTS		NKT							4

## Data Availability

The original contributions presented in this study are included in the article/Appendix A. Further inquiries can be directed to the corresponding authors.

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
