# Peer review of "Phylogenomic and Evolutionary Insights into Lipoprotein Lipase (LPL) Genes in Tambaqui: Gene Duplication, Tissue-Specific Expression and Physiological Implications"

_genes, 2025, doi:10.3390/genes16050548_

Round 1
Reviewer 1 Report
Comments and Suggestions for Authors
This work by Paixão et al., presents a detailed characterization of the lpl gene family in fish with a special focus on tambaqui (Colossoma macropomum). The authors provide distinct levels of analysis, from genome analysis, gene expression, to protein physicochemical property prediction. I find the study is generally sound, clear and well written. The study is descriptive and mostly inference-based, so the author should be conservative when implying functional variations and adaptive processes.
Line 133: To be accurate the phylogenetic analysis was done with LPL aa sequences
Line 202 (and results): The authors use “read depth” as a proxy to compare lpl expression across teleosts; read depth values were obtained from the Phylofish database. Although read depth relates with the number of reads that map to a specific gene, it also varies according to the sequencing depth (i.e. total number of reads per samples). Is the sequencing depth across species/tissues comparable?
Please standardize gene naming to improve readability: e.g. line 295 lpl and lpl-like and line 300 lpl1 and lpl2.
Line 341: “reflecting functional divergence and post-duplication adaptation”, although syntenic rearragement can affect gene regulation, I find the authors should be cafeful when directly affirming that rearrangement reflect functional divergence. The data is robust and adequate to infer orthologies, not function.
Line 410: correct Lp11a
Line 458: remove extra “.”
Line 462/654: the instability indexes and lenghtly describe but, what are the putative consequences of being stable/unstable? The author suggest “tailored to each species physiological and environmental needs”, could it be tissue dependent?
Figure 6: provide efficiency values for RT-PCR reations
Line 589: protein symbols not in italics
Line 612, Table S1, S2: Replace “basal teleost” by “Early branching teleost”
Line 631: The maintenance of syntenic blocks is more likely to be conserved between closely-related species, this does not necessarily reflect specific functional adaptations.
Line 699: Is there a relationship between glycosylation patterns in tambaqui and specific tissue expression/tissue environment (e.g. gonads vs liver)?
Author Response
Comment 1: Line 133: To be accurate the phylogenetic analysis was done with LPL aa sequences. Response1: We thank the reviewer for his/her kind comments. We agree with the reviewer and have revised the sentence for clarity and accuracy. The updated sentence (line 132-135) now reads: “This procedure produced multiple BLAST hits for LPL amino acid sequences in each of the protein databases, which were individually examined and retained in FASTA format for phylogenetic analysis.” Comment 2: Line 202 (and results): The authors use “read depth” as a proxy to compare lpl expression across teleosts; read depth values were obtained from the Phylofish database. Although read depth relates with the number of reads that map to a specific gene, it also varies according to the sequencing depth (i.e. total number of reads per sample). Is the sequencing depth across species/tissues comparable? Response 2: We thank the reviewer for your comments. We agree that sequencing depth can influence read depth values, which may limit direct quantitative comparisons across species or tissues if library construction and sequencing were not standardized. However, the PhyloFish database was specifically designed to enable interspecies comparisons of gene expression by using a consistent RNA-seq protocol across all 23 fish species, including (i) the same set of ten tissues, (ii) identical library preparation and sequencing chemistry, and (iii) comparable sequencing depth (Pasquier et al., 2016). These methodological controls minimize technical variability and allow mean depth values to be used as a reasonable proxy for cross-tissue expression patterns. In our study, we used average depth data primarily as an exploratory and qualitative reference to visualize the tissue-specific expression of lpl paralogs across teleosts. To clarify this point, we edited the sentence in line 211-215 as follows: Revised: “The average depth values for each lpl copy were obtained from the RNA-seq libraries available in the PhyloFish database (https://phylofish.sigenae.org/) using the TBLASTN algorithm with protein sequences as queries. These values were used qualitatively to visualize tissue-specific expression patterns, as sequencing depth across species and tissues was standardized in PhyloFish; nonetheless, differences in transcriptome completeness and expression dynamics may still limit direct quantitative comparisons (Supplementary Figure 1).” Comment 3: Please standardize gene naming to improve readability: e.g. line 295 lpl and lpl-like and line 300 lpl1 and lpl2. Response 3: Thank you for your observation. We have revised the manuscript to standardize gene nomenclature throughout the text, including line 296. The names lpl1, and lpl2 are now used consistently to improve clarity and readability. Comment 4: Line 341: “reflecting functional divergence and post-duplication adaptation”, although syntenic rearrangement can affect gene regulation, I find the authors should be careful when directly affirming that rearrangement reflects functional divergence. The data is robust and adequate to infer orthologies, not function. Response 4: We appreciate the reviewer’s valuable comment and fully agree that syntenic rearrangements alone are not sufficient to confirm functional divergence. Our intention was to suggest a possible link between the observed gene context variation and functional differentiation, not to assert it definitively. In light of this, we have revised the sentence for greater accuracy and to avoid overinterpretation. The revised sentence in lines Line 336-340 now reads: Revised: “The genes adjacent to lpl1b, tpm4, rab8a, cib3 (upstream), and ap1m1, klf2b, eps15l1, crt3, rx2 (downstream) are conserved in all teleosts that retained this copy, except for Mexican tetra, which shares only upstream genes in synteny. However, despite these variations, the syntenic organization of tpm4-lpl1b-ap1m1 is highly conserved among the teleost species analyzed” Comment 5: Line 410: correct Lp11a Response 5: Thank you for pointing that out. The typographical error in "Lp11a" has been corrected (Line 407) Comment 6: Line 458: remove extra “.” Response 6: Thank you for the observation. The extra period has been removed (Lines 224 and 451). Comment 7: Line 462/654: The instability indexes are described at length, but what are the putative consequences of being stable/unstable? The author suggests “tailored to each species physiological and environmental needs.” Could it be tissue-dependent? Response 7: We thank the reviewer for this insightful question. We agree that the implications of protein stability merit further clarification. To address this, we have expanded the discussion to briefly consider possible functional consequences (Lines 678-685). While the instability index is a predictive in silico parameter and does not necessarily reflect in vivo degradation rates, it can provide indirect clues about protein turnover, structural dynamics, or adaptation to specific cellular contexts. We have modified the manuscript as follows: Revised: “While the instability index is a theoretical prediction and does not directly determine in vivo protein degradation, lower stability may suggest a higher turnover rate, potentially affecting enzyme availability and activity under different physiological demands (Guruprasad et al., 1990). These variations may reflect specific adaptations in the optimal activity of lipases and temperature stability, tailored to each species physiological and environmental needs [60,61,62,63]. Additionally, such variation might also be tissue-dependent, as certain tissues could require more dynamic regulation of lipid metabolism than others (Weil et al., 2013).” Comment 8: Figure 6: provide efficiency values for RT-PCR reations Response 8: We thank the reviewer for this observation. We have now included the amplification efficiency values in the legend of Figure 6 and in the Materials and Methods section (lines 186–190). Revised: Amplification efficiency for each primer set was calculated from five-point, 1:4 serial dilution curves, using pooled liver cDNA for lpl1a and lpl2a, and pooled ovary and testis cDNA for lpl1b. The RT-qPCR assays showed high linearity and efficiency for all target genes, with amplification efficiencies of 101.39% (lpl1a), 97.75% (lpl1b), 95.99% (lpl2a), and 100.88% (β-actin).
Response 9: Thank you for pointing this out. We have carefully reviewed the manuscript and corrected the formatting to ensure that gene symbols are italicized and protein symbols are presented in regular font. Specifically, in line 578 (589 in the original document) and throughout the manuscript, we verified and standardized the use of italics and capitalization according to accepted gene and protein nomenclature guidelines. Comment 10: Line 612, Table S1, S2: Replace “basal teleost” by “Early branching teleost” Response 10: Done. The term "basal teleost" was replaced with "early branching teleost" throughout the manuscript and in Supplementary Tables S1 and S2 to ensure consistency and reflect current phylogenetic terminology. Comment 11: Line 631: The maintenance of syntenic blocks is more likely to be conserved between closely-related species, this does not necessarily reflect specific functional adaptations. Response 11: We appreciate the reviewer’s comment and agree that the conservation of syntenic blocks, especially among closely related species, should not be interpreted as direct evidence of functional adaptation. We have revised the text to adopt a more cautious interpretation, focusing on the potential for structural constraints and evolutionary pressures without making definitive claims about functional divergence (Lines 649-654). Comment 12: Line 699: Is there a relationship between glycosylation patterns in tambaqui and specific tissue expression/tissue environment (e.g. gonads vs liver)? Response 12: We thank the reviewer for this insightful question. We have incorporated a clarification in the revised manuscript to better link the glycosylation patterns of lpl1a and lpl1b with their tissue-specific expression profiles in tambaqui. Although the functional implications of glycosylation were initially discussed from a comparative and evolutionary perspective, we agree that tissue-specific expression data support a more targeted interpretation. In tambaqui, lpl1a is more highly expressed in the liver, while lpl1b is not detected in the liver but is predominantly expressed in the gonads. Given that Lpl1b contains more predicted N-glycosylation sites than Lpl1a, it is plausible that these differences reflect tissue-specific post-translational requirements, possibly related to the local microenvironment or physiological demands of lipid metabolism in the gonads. This connection has now been briefly mentioned in the glycosylation discussion and is further explored in the gene expression results section (Lines 779-787).
|

Reviewer 2 Report
Comments and Suggestions for Authors
The manuscript by Romulo Paixao et al., titled “Phylogenomic and Evolutionary Insights into Lipoprotein Lipase (LPL) Genes in Tambaqui: Gene Duplication, Tissue-Specific Expression, and Physiological Implications”, explores the presence and diversification of lpl gene copies in the teleost species tambaqui (Colossoma macropomum). The study presents phylogenetic analysis, including phylogenetic tree and synteny assessments, as well as a comparative evaluation of the deduced protein sequences. In addition, the authors examine the tissue-specific expression patterns of the identified lpl genes in tambaqui and compare these expression profiles with those of other teleost species.
Overview of the manuscript:
The introduction is well-constructed and, in this reviewer's opinion, appropriately outlines the current knowledge on the LPL gene and its function, particularly in mammals. It also briefly summarizes what is known in teleosts, providing a relevant context for the study. The Materials and Methods section is clearly written and enables the reader to follow the overall strategy used for the in-silico analyses, including phylogenetic, synteny, and protein structure assessments. The pipeline employed for the identification of paralogs and orthologs, particularly through phylogenetic tree construction and synteny analysis, is well designed and appropriate for the study's objectives. Additionally, the comparative protein analysis is thoroughly executed and adds valuable support to the evolutionary interpretation of the LPL gene family. The Results are generally well presented; however, some sentences would be more appropriate in the Discussion section. That said, there are several major issues that the authors are strongly encouraged to address (detailed below).
This reviewer suggestion: Major revision
The following are the key concerns that the authors are strongly encouraged to address in a revised version of the manuscript:
- Gene Nomenclature
Gene naming—especially when proposing changes—should follow established nomenclature guidelines to maintain consistency across species. Renaming the lpl gene (an ortholog of human LPL) to lpl1, as proposed in this manuscript, may disrupt cross-species comparability and create unnecessary confusion. Furthermore, the gene referred to as lpl-like (renamed lpl2 in the manuscript) shares structural similarities with lpl, but its function remains uncharacterized. The absence of a clear ortholog in tetrapods further complicates any functional inference, making it premature to rename this gene lpl2 without additional functional evidence.
In this reviewer’s opinion, assigning the name lpl to a gene implies functional conservation with its mammalian counterpart, which may not be appropriate in this case. Additionally, if the authors wish to rename lpl to lpl1, they should have submitted a request for nomenclature approval to ZFIN (Zebrafish Nomenclature Conventions), as outlined in the guidelines:
🔗 https://zfin.atlassian.net/wiki/spaces/general/pages/1818394635/ZFIN+Zebrafish+Nomenclature+Conventions
Such approval would provide greater clarity, ensure community-wide consistency, and offer more confidence in accepting the proposed nomenclature changes.
- qPCR Normalization
The authors report using only one reference gene for the qPCR expression analysis. However, it is widely accepted that using at least two validated reference genes is necessary to ensure accurate normalization and reliable results. Relying on a single reference gene—especially one whose stability has not been rigorously confirmed—can lead to misleading interpretations, as expression levels may vary depending on tissue type, developmental stage, or external conditions. Best practices typically involve selecting a panel of candidate reference genes and then evaluating their stability using established algorithms (such as geNorm or/and NormFinder) to identify the most appropriate housekeeping genes for the specific experimental context.
Thus, in this reviewer’s opinion, the expression analysis and normalization of LPL transcripts in the current manuscript require further validation. Incorporating multiple, validated reference genes would significantly strengthen the accuracy and credibility of the qPCR-based findings.
Minor Comments
- It is recommended that the third and fourth whole-genome duplication (WGD) events be visually distinguished in the phylogenetic tree using different colors. This would improve clarity and help readers more easily interpret the evolutionary context.
- As previously mentioned, some sentences in the Results section go beyond reporting findings and begin to interpret or discuss the data. These sentences would be more appropriately placed in the Discussion section to maintain a clear separation between results and interpretation.
- Given the large number of sequences used in the in-silico analyses, it would be helpful to include additional details in the supplementary table. Specifically, where possible, please indicate the chromosomal location of each gene. This information would provide stronger support for the hypothesis that the identified paralogs arose from WGD events.
Author Response
Gene naming—especially when proposing changes—should follow established nomenclature guidelines to maintain consistency across species. Renaming the lpl gene (an ortholog of human LPL) to lpl1, as proposed in this manuscript, may disrupt cross-species comparability and create unnecessary confusion. Furthermore, the gene referred to as lpl-like (renamed lpl2 in the manuscript) shares structural similarities with lpl, but its function remains uncharacterized. The absence of a clear ortholog in tetrapods further complicates any functional inference, making it premature to rename this gene lpl2 without additional functional evidence. In this reviewer’s opinion, assigning the name lpl to a gene implies functional conservation with its mammalian counterpart, which may not be appropriate in this case. Additionally, if the authors wish to rename lpl to lpl1, they should have submitted a request for nomenclature approval to ZFIN (Zebrafish Nomenclature Conventions), as outlined in the guidelines: Such approval would provide greater clarity, ensure community-wide consistency, and offer more confidence in accepting the proposed nomenclature changes. In our study, we refer to lpl1 and lpl2 to distinguish two paralogous genes found in teleosts and other non-tetrapod vertebrates. These genes do not result from the teleost-specific whole genome duplication, but rather originate from an ancient tandem duplication that predates the divergence between sarcopterygians and actinopterygians. This inference is supported by our phylogenetic analyses, which place lpl1 and lpl2 in well-supported and distinct clades, with conserved synteny observed in multiple vertebrate lineages. The nomenclature lpl1 and lpl2 has precedent in previous studies in fish species such as medaka, red seabream, and Fugu (e.g., Wang et al., 2015; Oku et al., 2006; Kaneko et al., 2013), where both copies are retained and differentially expressed. In our manuscript, we adopted these names to facilitate evolutionary and comparative genomic analyses and to maintain consistency with established literature. We emphasize that lpl1 corresponds to the ortholog of the single LPL gene found in mammals, whereas lpl2 refers to its paralog that was retained in specific lineages following the ancestral duplication. We acknowledge that ZFIN guidelines discourage the use of suffixes such as “a/b” or “.1/.2” when duplications predate the ray-finned/lobe-finned divergence. However, given the absence of a one-to-one orthologous relationship to mammalian LPL in species where both copies are present, and the importance of distinguishing between these paralogs, we opted for the lpl1/lpl2 designation. This usage is not intended to imply functional equivalence with mammalian LPL or to represent a formal nomenclature proposal. To avoid confusion, we have revised the manuscript to explicitly state that these gene names are used only to distinguish ancient paralogs and that functional divergence of lpl2 remains to be experimentally validated. We have also removed any language suggesting formal gene symbol proposals and clarified our intent to submit a nomenclature request to ZFIN in the future should further functional and comparative evidence support such an action. We believe that these revisions strengthen the manuscript by improving clarity while maintaining a biologically meaningful distinction between the lpl paralogs in teleosts and other vertebrates.
Thus, in this reviewer’s opinion, the expression analysis and normalization of LPL transcripts in the current manuscript require further validation. Incorporating multiple, validated reference genes would significantly strengthen the accuracy and credibility of the qPCR-based findings.
Response 2: We appreciate the reviewer’s important comment regarding reference gene selection for qPCR normalization. We fully agree that the use of multiple validated reference genes is considered best practice, particularly when analyzing gene expression across diverse tissues and conditions. Although β-actin was used as the sole reference gene in our qPCR analyses, its suitability for normalization in Colossoma macropomum has been previously validated. Nascimento et al. (2016) evaluated the expression stability of several candidate reference genes and identified β-actin as the most stable, followed by 18S rRNA. Nevertheless, we recognize that optimal qPCR normalization typically requires the use of multiple reference genes validated for each experimental condition. Therefore, our current expression screening should be interpreted as qualitative, aiming to detect presence/absence and tissue-level patterns of expression rather than provide absolute quantification. Future studies will incorporate multiple reference genes and stability analysis tools (e.g., geNorm, NormFinder) to enhance the accuracy and reliability of gene expression quantification.
References: ● Gallani, S. U., Michelato, M., Vargas, L. et al. (2021). Expression of immune-related genes in spleen and kidney of tambaqui (Colossoma macropomum) fed with dietary probiotics and challenged with Aeromonas hydrophila. Microbial Pathogenesis, 150:104638. https://doi.org/10.1016/j.micpath.2020.104638 ● Feng, H., Guo, S., Wang, L. et al. (2014). Molecular cloning and expression analysis of lipoprotein lipase gene in zebrafish (Danio rerio). Journal of Fish Biology, 84(3): 830–844. https://doi.org/10.1111/jfb.12423 ● Wang, Z., Zhu, Q., Yang, H. et al. (2015). Molecular cloning and expression analysis of lipoprotein lipase gene in medaka (Oryzias latipes). Fisheries Science, 81: 497–506. https://doi.org/10.1007/s12562-014-0826-7 ● Oku, H., Tsuchioka, M., Yamaguchi, K., & Ando, M. (2006). Cloning and expression of lipoprotein lipase and hepatic lipase in red sea bream (Pagrus major). Comparative Biochemistry and Physiology Part B: Biochemistry and Molecular Biology, 145(3–4), 345–355. https://doi.org/10.1016/j.cbpb.2006.06.008 ● Kaneko et al., 2013
Response 3: We thank the reviewer for this helpful suggestion. We have now updated the phylogenetic tree by using different colors to visually distinguish the third (3R) and fourth (4R) whole-genome duplication events, thus improving the clarity and interpretability of the evolutionary context.
Comment 4: As previously mentioned, some sentences in the Results section go beyond reporting findings and begin to interpret or discuss the data. These sentences would be more appropriately placed in the Discussion section to maintain a clear separation between results and interpretation.
Response 5: Thank you for the suggestion. We have updated the supplementary Figure S1 to include the chromosomal location of each gene, where available. This additional information helps strengthen the evidence for the proposed whole-genome duplication (WGD) origin of the identified paralogs. |
|
|
|
4. Response to Comments on the Quality of English Language |
Point 1: The English is fine and does not require any improvement. |
Response 1: Thank you for your feedback. We appreciate your assessment of the language quality and are glad to know that no further improvements are necessary.
|
